# BaB-prob: Branch and Bound with Preactivation Splitting for Probabilistic Verification of Neural Networks

## Abstract

Branch-and-bound with preactivation splitting has been shown highly effective for deterministic verification of neural networks. In this paper, we extend this framework to the probabilistic setting. We propose BaB-prob that iteratively divides the original problem into subproblems by splitting preactivations and leverages linear bounds computed by linear bound propagation to bound the probability for each subproblem. We prove soundness and completeness of BaB-prob for feedforward-ReLU neural networks. Furthermore, we introduce the notion of uncertainty level and design two efficient strategies for preactivation splitting, yielding BaB-prob-ordered and BaB+BaBSR-prob. We evaluate BaB-prob on untrained networks, MNIST and CIFAR-10 models, respectively, and VNN-COMP 2025 benchmarks. Across these settings, our approach consistently outperforms state-of-the-art approaches in medium- to high-dimensional input problems.

## 1 Introduction

*Probabilistic verification of neural networks* asks whether a given satisfies a formal specification with high probability under a prescribed input distribution. Formally, given a neural network $f\colon \mathbb{R}^n \to \mathbb{R}^m$, a random input $\mathbf{X} \in \mathbb{R}^n$ following distribution $\mathbf{P}$, a specification set $\mathcal{Y} \subset \mathbb{R}^m$, and a desired probability threshold $\eta \in (0, 1]$, the goal of probabilistic verification is to determine whether

$$\mathbb{P}\left(f(\mathbf{X}) \in \mathcal{Y}\right) \geq \eta \tag{1}$$

holds. In this paper, we consider the problem where $f = g^{(N)} \circ \sigma^{(N-1)} \circ \cdots \circ \sigma^{(1)} \circ g^{(1)}\colon \mathbb{R}^n \to \mathbb{R}$ is a feedforward-ReLU neural network, $g^{(1)}, \ldots, g^{(N)}$ are linear layers and $\sigma^{(1)}, \ldots, \sigma^{(N-1)}$ are ReLU layers, given as $\sigma_{\mathrm{ReLU}}(x) = \max(0, x)$, and $\mathcal{Y} = \{\mathbf{y} \in \mathbb{R}^m \colon \mathbf{c}^\mathsf{T}\mathbf{y} + d > 0\}$ is a half-space. By folding the $\mathbf{c}^\mathsf{T}\mathbf{y} + d$ into the last linear layer to obtain an equivalent scalar network $f\colon \mathbb{R}^n \to \mathbb{R}$, we finally formulate our probabilistic verification problem as follows:

**Problem 1.** *Determine whether the following statement is TRUE:*

$$\mathbb{P}\left(f(\mathbf{X}) > 0\right) \geq \eta, \tag{2}$$

*where $f = g^{(N)} \circ \sigma^{(N-1)} \circ \cdots \circ \sigma^{(1)} \circ g^{(1)}\colon \mathbb{R}^n \to \mathbb{R}$ is a feedforward-ReLU neural network, $\mathbf{X} \sim \mathbf{P}$ is a random input in $\mathbb{R}^n$, $\eta \in (0, 1]$ is the desired probability threshold.*

A verification approach is sound if, whenever it declares Problem 1 to be TRUE, then Problem 1 is indeed TRUE; it is complete if it declares Problem 1 to be TRUE whenever Problem 1 is TRUE.

Existing approaches for probabilistic verification can be divided into two categories: analytical approaches and sampling-based approaches (Sivaramakrishnan et al. (2024)). Sampling-based approaches sample a number of elements from the distribution, pass them through the neural network, then output a statistical result for the probabilistic verification problem with a specified confidence. A major advantage of sampling-based approaches is that they can be applied to arbitrary neural networks. However, the required number of samples increases with a higher confidence. For further details about sampling-based approaches, the reader is referred to Anderson & Sojoudi (2022); Devonport & Arcak (2020); Mangal et al. (2019); Pautov et al. (2022); Sivaramakrishnan et al. (2024).

In this paper, we mainly focus on analytical approaches. Analytical approaches can deterministically ascertain the correctness of Problem 1. Pilipovsky et al. (2023) compute the distribution of $f(\mathbf{X})$ using characteristic functions. However, their approach assumes independence between neurons at each layer. Besides, their approach is computationally impractical for deep and wide neural networks. PROVEN (Weng et al. (2019)) enjoys the best scalability among all current analytical methods. In particular, PROVEN leverages linear bound propagation based approaches (Zhang et al. (2018); Xu et al. (2020); Wang et al. (2018)) to get linear bounds on $f(\mathbf{x})$ and bound the probability in Equation (2) using these linear bounds. PROVEN is sound but incomplete due to the relaxations. Fazlyab et al. (2019) relaxes the original problem to a semidefinite programming problem by abstracting the nonlinear activation functions by affine and quadratic constraints. This approach is also sound but incomplete and lacks scalability to medium-size or large neural networks.

Branch and bound (BaB) is a technique widely used in deterministic verification (Bunel et al. (2020); Wang et al. (2021); Zhang et al. (2022)). BaB iteratively divides the problem into subproblems and solves individually. The subproblems are typically solved with tighter relaxations than the original problem under the constraints of the subproblems. After a finite number of splits, BaB-based approaches generate sound and complete results. BaB-based approaches for deterministic verification typically use one of two strategies: input-space splitting and preactivation splitting. Existing works (Boetius et al.; Marzari et al. (2024)) that use BaB for probabilistic verification focus on input-space splitting. However, as suggested in Bunel et al. (2020), preactivation splitting is considerably more efficient than input-space splitting for large networks, especially when the input space is high-dimensional.

In this work, we extend BaB with preactivation splitting to probabilistic verification. It should be noted that our approach can be used for general activation functions $\sigma^{(k)}$, however, we only prove completeness for ReLU activation function, given as $\sigma_{\text{ReLU}}(x) = \max(0, x)$. The proof can be extended to general piecewise-linear activation functions.

Our main contributions are as follows:

- We propose BaB-prob, the first branch-and-bound with preactivation splitting approach for probabilistic verification of neural networks.

- We prove the soundness and the completeness of BaB-prob for feedforward-ReLU networks, which also adapts to general piecewise linear activation functions.

- We introduce the concept of uncertainty level and leverage it to design two splitting strategies, yielding BaB-prob-ordered and BaB+BaBSR-prob.

- Experimental results show that BaB-prob-ordered and BaB+BaBSR-prob significantly outperform state-of-the-art probabilistic verification approaches when the input is medium-to high-dimensional.

## 2 PRELIMINARIES

In this section, we first introduce some notation used in the remainder of the paper. Then, we outline the linear bound propagation technique for deterministic verification used later in the proposed approach.

### 2.1 NOTATION

We denote the number of neurons of layer $g^{(k)}$ and layer $\sigma^{(k)}$ as $n_k$, $k = 1, \ldots, N-1$, and define $n_0 := n$ and $n_N := 1$. For $k = 1, \ldots, N-1$, we define $\mathbf{y}^{(k)}$ as the *preactivation* of layer $\sigma^{(k)}$ and $\mathbf{y}^{(k)}(\cdot) := g^{(k)} \circ \sigma^{(k-1)} \circ \cdots \circ g^{(1)}(\cdot)$ as the *preactivation function* of layer $\sigma^{(k)}$. For uniformity, we sometimes also use $\mathbf{y}^{(N)}(\cdot)$ to denote $f(\cdot)$. We use the shorthand notation $[k_1, k_2]$, where $k_1 \leq k_2$ to denote the sequence $\{k_1, k_1 + 1, \ldots, k_2\}$. We also use $[k]$ as shorthand for $[1, k]$ and $a^{[k]}$ for the sequence $\{a^{(1)}, a^{(2)}, \ldots, a^{(k)}\}$. We use the bold font $\mathbf{v}$ to denote a vector, and the regular font with subscript $v_j$ to denote its $j$-th entry. For two vectors $\mathbf{u}$ and $\mathbf{v}$ in $\mathbb{R}^{\ell}$, we denote $\mathbf{u} \leq \mathbf{v}$ if $u_i \leq v_i$ for all $i \in [\ell]$.

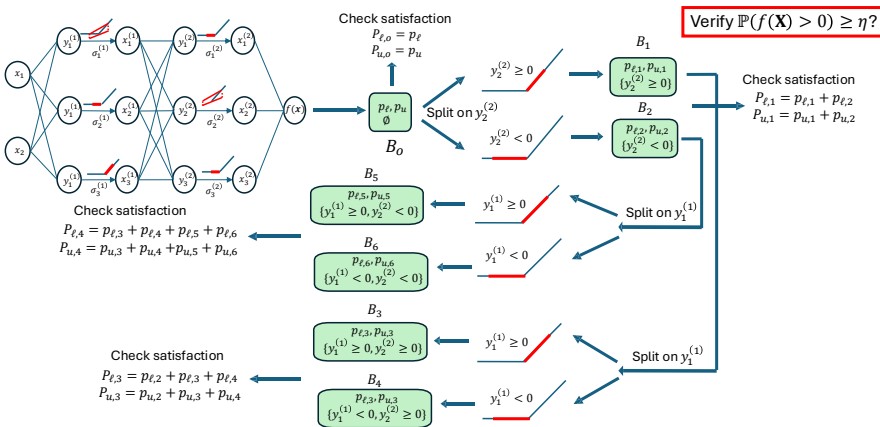

Figure 1: An illustrative example of the key idea behind BaB-prob. Two preactivations have negative lower bounds and positive upper bounds. BaB-prob iteratively splits them into negative and nonnegative cases to tighten the global probability bound, until the original problem is verified.

## 2.2 LINEAR BOUND PROPAGATION FOR DETERMINISTIC VERIFICATION

Different from verifying whether $f(\mathbf{X}) > 0$ is satisfied with high probability, deterministic verification verifies whether $f(\mathbf{x}) > 0$ for all $\mathbf{x}$ in a given input set $\mathcal{D}$. Formal verification is NP-hard due to its nonlinearity and nonconvexity (Katz et al. (2017)). To address this, linear bound propagation relaxes the problem by replacing the nonlinearities of $f$ by linear functions. Specifically, let $\mathbf{l}^{[N-1]}$ and $\mathbf{u}^{[N-1]}$ be precomputed lower and upper bounds for $\mathbf{y}^{[N-1]}$ such that $\ell_j^{(k)} \leq y_j^{(k)}(\mathbf{x}) \leq u_j^{(k)}$ for all $\mathbf{x} \in \mathcal{D}$ and $k \in [N-1]$. The precomputed bounds can be obtained via interval arithmetic or by applying linear bound propagation in the same way as for $f(\mathbf{x})$. Using the precomputed bounds, the ReLU functions can be relaxed by linear lower and upper functions. Relaxation is not required if $\ell_j^{(k)} \geq 0$ or $u_j^{(k)} \leq 0$ for some $y_j^{(k)}$. With these linear relaxations, $f(\mathbf{x})$ can be bounded by backward propagating through the layers, yielding linear lower and upper bounds, $\underline{f}(\mathbf{x})$ and $\bar{f}(\mathbf{x})$, on $f(\mathbf{x})$, satisfying $\underline{f}(\mathbf{x}) \leq f(\mathbf{x}) \leq \bar{f}(\mathbf{x})$, for all $\mathbf{x} \in \mathcal{D}$. It then verifies the problem using the linear bounds.

We can also leverage linear bound propagation to compute linear bounds for preactivation functions. The linear bounds for both of $f(\mathbf{x})$ and the preactivation functions will be used later in the proposed approach. A key advantage of linear bound propagation is its high degree of parallelism, making it well-suited for GPU acceleration. For more details about linear bound propagation, readers are referred to Zhang et al. (2018).

## 3 BAB-PROB

We now present our BaB-based approach for probabilistic verification, which we call BaB-prob. BaB-prob iteratively partitions Problem 1 into two subproblems by splitting a preactivation into negative and nonnegative cases, thereby rendering the ReLU function linear within each subproblem. It then applies linear bound propagation to compute linear bounds for $f$ and the preactivations $\mathbf{y}^{[N-1]}$. Using these bounds, BaB-prob computes probability bounds for each subproblem and aggregates them to obtain a global probability bound for the original problem. This process continues until Problem 1 is certified as either TRUE or FALSE. Figure 1 provides an example illustrating this key idea.

The remainder of this section is organized as follows: Section 3.1 outlines the overall framework of BaB-prob; Section 3.2 explains a key component—bounding the probability of a branch; Section 3.3 proposes two strategies for selecting preactivations to split on based on the concept of uncertainty level. The pseudocode and further implementation details of the framework are provided in Appendix A.

### 3.1 Overall Framework

A *set of constraints* is of the form $\mathcal{C} = \big\{ y_{j_1}^{(k_1)} \geq 0, \ldots, y_{j_s}^{(k_s)} \geq 0, y_{j_{s+1}}^{(k_{s+1})} < 0, \ldots y_{j_t}^{(k_t)} < 0 \big\}$. We say that $\mathcal{C}$ is *satisfied* if all the constraints in $\mathcal{C}$ hold simultaneously. We can decompose $\mathcal{C}$ layer by layer as $\mathcal{C} = \mathcal{C}^{(1)} \cup \cdots \cup \mathcal{C}^{(N-1)}$, where $\mathcal{C}^{(k)}$ denotes the set of constraints on $\mathbf{y}^{(k)}$ and is the empty set when there is no constraint on $\mathbf{y}^{(k)}$. When $\mathcal{C}$ appears inside a probability, $\mathbb{P}(\cdot)$, it denotes the event that all the constraints in $\mathcal{C}$ are satisfied. For instance, $\mathbb{P}(\mathcal{C}) = \mathbb{P}\left( y_{j_1}^{(k_1)}(\mathbf{X}) \geq 0, \ldots, y_{j_s}^{(k_s)}(\mathbf{X}) \geq 0, y_{j_{s+1}}^{(k_{s+1})}(\mathbf{X}) < 0, \ldots, y_{j_t}^{(k_t)}(\mathbf{X}) < 0 \right)$. A *branch* is defined as $B := \langle p_\ell, p_u, \mathcal{C} \rangle$, where $\mathcal{C}$ is the set of constraints for $B$, and $p_\ell$ and $p_u$ are lower and upper bounds on $\mathbb{P}(f(\mathbf{X}) > 0, \mathcal{C})$, respectively. We define the *probability* of the branch $B$ as $\mathbb{P}(B) := \mathbb{P}(f(\mathbf{X}) > 0, \mathcal{C})$. For a preactivation $y_j^{(k)}$ with precomputed lower bound $\ell_j^{(k)}$ and upper bound $u_j^{(k)}$, we say that the preactivation is *stable* in $B = \langle p_\ell, p_u, \mathcal{C} \rangle$ if $\ell_j^{(k)} \geq 0$, or $u_j^{(k)} \leq 0$, or $\mathcal{C}$ imposes a constraint on $y_j^{(k)}$; otherwise, we say that it is *unstable* in $B$. We assume $\mathbf{X}$ is bounded in $\mathcal{D}$; if not, take $\mathcal{D}$ as a $(1 - \delta)$-confidence set for $\mathbf{X}$.

During initialization, $\mathcal{B}$ is created to store all candidate branches. BaB-prob then performs linear bound propagation over $\mathcal{D}$ **under no constraint** to obtain linear bounds for $f$ and $\mathbf{y}^{[N-1]}$, from which the probability $\mathbb{P}(f(\mathbf{X}) > 0)$ can be bounded between $p_\ell$ and $p_u$. The root branch, $B_o := \langle p_\ell, p_u, \mathcal{C} = \varnothing \rangle$, is then inserted into $\mathcal{B}$. At each iteration, BaB-prob first aggregates across branches in $\mathcal{B}$ to compute the global probability bounds and terminates if Problem 1 is already certified with the current bounds. Otherwise it pops the branch $B = \langle p_\ell, p_u, \mathcal{C} \rangle$ with largest gap $(p_u - p_\ell)$. The gap must be positive, since otherwise the global lower and upper bounds would coincide and BaB-prob would have already terminated. It then picks an unstable preactivation $y_j^{(k)}$ in $B$, and creates two new sets of constraints $\mathcal{C}_1 = \mathcal{C} \cup \{y_j^{(k)} \geq 0\}$ and $\mathcal{C}_2 = \mathcal{C} \cup \{y_j^{(k)} < 0\}$. Applying linear bound propagation over $\mathcal{D}$ **under the constraints in** $\mathcal{C}_i$, BaB-prob recomputes linear bounds for $f$ and $\mathbf{y}^{[N-1]}$, yielding $\underline{f}_i(\mathbf{x})$, $\bar{f}_i(\mathbf{x})$ and $\underline{\mathbf{y}}_i^{[N-1]}(\mathbf{x})$, $\bar{\mathbf{y}}_i^{[N-1]}$, such that

$$\underline{f}_i(\mathbf{x}) \leq f(\mathbf{x}) \leq \bar{f}_i(\mathbf{x}), \qquad \forall \mathbf{x} \in \mathcal{D}, \ \mathcal{C}_i \text{ satisfied}, \tag{3a}$$

$$\underline{\mathbf{y}}_i^{(k)}(\mathbf{x}) \leq \mathbf{y}^{(k)}(\mathbf{x}) \leq \bar{\mathbf{y}}_i^{(k)}(\mathbf{x}), \qquad \forall \mathbf{x} \in \mathcal{D}, \ \mathcal{C}_i^{[k-1]} \text{ satisfied}, \ k \in [N-1]. \tag{3b}$$

With the linear bounds, it computes probability bounds $p_\ell$ and $p_u$ for $\mathbb{P}(f(\mathbf{X}) > 0, \mathcal{C}_i)$. It then inserts $B_i := \langle p_{\ell,i}, p_{u,i}, \mathcal{C}_i \rangle$ to $\mathcal{B}$, and continues.

Corollary 1 shows that BaB-prob is both sound and complete.

### 3.2 Probability Bounds for Branches

Let $B$ be a given branch, and $\mathcal{C} = \big\{ y_{j_1}^{(k_1)} \geq 0, \ldots, y_{j_s}^{(k_s)} \geq 0, y_{j_{s+1}}^{(k_{s+1})} < 0, \ldots y_{j_t}^{(k_t)} < 0 \big\}$ be the set of constraints for it. Assume the linear bounds for $f$ and $\mathbf{y}^{[N-1]}$ under the constraints in $\mathcal{C}$ are $\underline{f}(\mathbf{x}) = \underline{\mathbf{a}}^\mathsf{T}\mathbf{x} + \underline{b}$, $\bar{f}(\mathbf{x}) = \bar{\mathbf{a}}^\mathsf{T}\mathbf{x} + \bar{b}$, $\underline{\mathbf{y}}^{(k)}(\mathbf{x}) = \underline{\mathbf{A}}^{(k)}\mathbf{x} + \underline{\mathbf{b}}^{(k)}$, $\bar{\mathbf{y}}^{(k)}(\mathbf{x}) = \bar{\mathbf{A}}^{(k)}\mathbf{x} + \bar{\mathbf{b}}^{(k)}$, for $k \in [N-1]$. Then, the probability bounds for $B$ are given by

$$p_\ell = \mathbb{P}\left( \underline{\mathbf{P}}\mathbf{X} + \underline{\mathbf{q}} \leq 0 \right), \quad p_u = \mathbb{P}\left( \bar{\mathbf{P}}\mathbf{X} + \bar{\mathbf{q}} \leq 0 \right), \tag{4}$$

where,

$$\underline{\mathbf{P}} = \begin{pmatrix} -\underline{\mathbf{a}}^\mathsf{T} \\ -\underline{\mathbf{A}}_{j_1,:}^{(k_1)} \\ \vdots \\ -\underline{\mathbf{A}}_{j_s,:}^{(k_s)} \\ \bar{\mathbf{A}}_{j_{s+1},:}^{(k_{s+1})} \\ \vdots \\ \bar{\mathbf{A}}_{j_t,:}^{(k_t)} \end{pmatrix}, \quad \underline{\mathbf{q}} = \begin{pmatrix} -\underline{b} \\ -\underline{b}_{j_1}^{(k_1)} \\ \vdots \\ -\underline{b}_{j_s}^{(k_s)} \\ \bar{b}_{j_{s+1}}^{(k_{s+1})} \\ \vdots \\ \bar{b}_{j_t}^{(k_t)} \end{pmatrix}, \quad \bar{\mathbf{P}} = \begin{pmatrix} -\bar{\mathbf{a}}^\mathsf{T} \\ -\bar{\mathbf{A}}_{j_1,:}^{(k_1)} \\ \vdots \\ -\bar{\mathbf{A}}_{j_s,:}^{(k_s)} \\ \underline{\mathbf{A}}_{j_{s+1},:}^{(k_{s+1})} \\ \vdots \\ \underline{\mathbf{A}}_{j_t,:}^{(k_t)} \end{pmatrix}, \quad \bar{\mathbf{q}} = \begin{pmatrix} -\bar{b} \\ -\bar{b}_{j_1}^{(k_1)} \\ \vdots \\ -\bar{b}_{j_s}^{(k_s)} \\ \underline{b}_{j_{s+1}}^{(k_{s+1})} \\ \vdots \\ \underline{b}_{j_t}^{(k_t)} \end{pmatrix}.$$

It is proved in Proposition 1 that $p_\ell$ and $p_u$ in Equation (4) are indeed lower and upper bounds on $\mathbb{P}\left(f(\mathbf{X}) > 0, \mathcal{C}\right)$. Note that Equation (4) corresponds to the cumulative density functions of linear transformations of $\mathbf{X}$ at the origin. If $\mathbf{P}$ is Gaussian, Equation (4) can be computed easily since the linear transformation of a Gaussian random variable variable is still Gaussian. If $\mathbf{P}$ is a general distribution, Equation (4) can be computed by integrating the probability density functions of $\mathbf{X}$ or using Monte Carlo sampling.

### 3.3 SPLITTING STRATEGIES

We first derive the *uncertainty levels* of unstable preactivations in a branch $B$, which reflects the looseness of the probability bounds of $B$ that Equation (4) will provide after splitting on the preactivations. Then, we propose two splitting strategies based on uncertainty level.

**Uncertainty level.** Consider a branch $B = \langle p_\ell, p_u, \mathcal{C} \rangle$, where $\mathcal{C} = \{y_{j_1}^{(k_1)} \geq 0, \ldots, y_{j_s}^{(k_s)} \geq 0, y_{j_{s+1}}^{(k_{s+1})} < 0, \ldots, y_{j_t}^{(k_t)} < 0\}$. Let $y_j^{(k)}$ be an unstable preactivation in $B$. Let $B_1 = \langle p_{\ell,1}, p_{u,1}, \mathcal{C}_1 \rangle$ and $B_2 = \langle p_{\ell,2}, p_{u,2}, \mathcal{C}_2 \rangle$ be the children branches generated if we split on $y_j^{(k)}$, where $\mathcal{C}_1 = \mathcal{C} \cup \{y_j^{(k)} \geq 0\}$ and $\mathcal{C}_2 = \mathcal{C} \cup \{y_j^{(k)} < 0\}$. Assume $\underline{f}_1(x)$, $\bar{f}_1(\mathbf{x})$, $\underline{\mathbf{y}}_1^{[N-1]}(\mathbf{x})$, $\bar{\mathbf{y}}_1^{[N-1]}(\mathbf{x})$ and $\underline{f}_2(\mathbf{x})$, $\bar{f}_2(\mathbf{x})$ $\underline{\mathbf{y}}_2^{[N-1]}(\mathbf{x})$, $\bar{\mathbf{y}}_2^{[N-1]}(\mathbf{x})$ are the linear upper and lower bounds for $f$ and $\mathbf{y}^{[N-1]}$ obtained by linear bound propagation under the constraints in $\mathcal{C}_1$ and $\mathcal{C}_2$, respectively.

By Proposition 2,

$$
\begin{aligned}
p_{u,1} - p_{\ell,1} \quad &\geq \mathbb{P} \begin{pmatrix} \bar{y}_{j,1}^{(k)}(\mathbf{X}) \geq 0, \underline{y}_{j,1}^{(k)}(\mathbf{X}) < 0, \bar{f}_1(\mathbf{X}) > 0, \\ \bar{y}_{j_1,1}^{(k_1)}(\mathbf{X}) \geq 0, \ldots, \bar{y}_{j_s,1}^{(k_s)}(\mathbf{X}) \geq 0, \underline{y}_{j_{s+1},1}^{(k_{s+1})}(\mathbf{X}) < 0, \ldots, \underline{y}_{j_t,1}^{(k_t)}(\mathbf{X}) < 0 \end{pmatrix} \\
&=: q(B_1), \quad\quad\quad\quad\quad\quad\quad\quad\quad\quad\quad\quad\quad\quad\quad\quad\quad\quad\quad\quad\quad\quad\quad\quad (5a) \\
p_{u,2} - p_{\ell,2} \quad &\geq \mathbb{P} \begin{pmatrix} \bar{y}_{j,2}^{(k)}(\mathbf{X}) \geq 0, \underline{y}_{j,2}^{(k)}(\mathbf{X}) < 0, \bar{f}_2(\mathbf{X}) > 0, \\ \bar{y}_{j_1,2}^{(k_1)}(\mathbf{X}) \geq 0, \ldots, \bar{y}_{j_s,2}^{(k_s)}(\mathbf{X}) \geq 0, \underline{y}_{j_{s+1},2}^{(k_{s+1})}(\mathbf{X}) < 0, \ldots, \underline{y}_{j_t,2}^{(k_t)}(\mathbf{X}) < 0 \end{pmatrix} \\
&=: q(B_2). \quad\quad\quad\quad\quad\quad\quad\quad\quad\quad\quad\quad\quad\quad\quad\quad\quad\quad\quad\quad\quad\quad\quad\quad (5b)
\end{aligned}
$$

Equation (5) implies, in particular, that there are gaps, denoted by $q(B_1)$ and $q(B_2)$, between the lower and upper bounds on $\mathbb{P}(B_1)$ and $\mathbb{P}(B_2)$. Note that these gaps arise from the relaxation inherent in our approach. If no relaxation were used when computing $\mathbb{P}(B_1)$ and $\mathbb{P}(B_2)$, the gaps would vanish. A natural idea is to split on the unstable preactivation that will result in smallest gap. However, computing the gap exactly for every unstable neuron is computationally intractable: it requires computing the linear bounds of $f$ and $\mathbf{y}^{[N-1]}$ under the corresponding constraints, together with the probabilities on the right-hand side of Equation (5), for each unstable preactivation. Instead, we use upper bounds on $q(B_1)$ and $q(B_2)$, denoted by $\bar{q}(B_1)$ and $\bar{q}(B_2)$, which can be computed more easily. The upper bounds are given by

$$
q(B_1) \leq \mathbb{P}\left(\bar{y}_{j,1}^{(k)}(\mathbf{X}) \geq 0, \underline{y}_{j,1}^{(k)}(\mathbf{X}) < 0\right) =: \bar{q}(B_1), \tag{6a}
$$

$$
q(B_2) \leq \mathbb{P}\left(\bar{y}_{j,2}^{(k)}(\mathbf{X}) \geq 0, \underline{y}_{j,2}^{(k)}(\mathbf{X}) < 0\right) =: \bar{q}(B_2). \tag{6b}
$$

In many linear bound propagation based approaches, such as CROWN (Zhang et al. (2018)), the computed linear lower and upper bounds for $y_j^{(k)}(\mathbf{x})$ are determined by the constraints on the preactivations of the first $k - 1$ ReLU layers. Therefore, and since $\mathcal{C}_1$ and $\mathcal{C}_2$ have the same constraints as $\mathcal{C}$ on $\mathbf{y}^{[k-1]}$, the linear lower and upper bounds on $y_j^{(k)}(\mathbf{x})$ for $B_1$ and $B_2$ are same as those for $B$, that is,

$$
\underline{y}_{j,1}^{(k)}(\mathbf{x}) = \underline{y}_{j,2}^{(k)}(\mathbf{x}) = \underline{y}_j^{(k)}(\mathbf{x}), \quad \bar{y}_{j,1}^{(k)}(\mathbf{x}) = \bar{y}_{j,2}^{(k)}(\mathbf{x}) = \bar{y}_j^{(k)}(\mathbf{x}). \tag{7}
$$

Therefore, $\bar{q}(B_1) = \bar{q}(B_2)$, and we denote them by the *uncertainty level* of unstable preactivation $y_j^{(k)}$ in branch $B$, that is,

$$
q(B; y_j^{(k)}) := \mathbb{P}\left(\bar{y}_j^{(k)}(\mathbf{X}) \geq 0, \underline{y}_j^{(k)}(\mathbf{X}) < 0\right) = \mathbb{P}\left(\begin{pmatrix} -\bar{\mathbf{A}}_{j,:}^{(k)} \\ \underline{\mathbf{A}}_{j,:}^{(k)} \end{pmatrix} \mathbf{X} + \begin{pmatrix} -\bar{b}_j^{(k)} \\ \underline{b}_j^{(k)} \end{pmatrix} \leq 0\right), \tag{8}
$$

where $\underline{y}_j^{(k)}(\mathbf{x}) = \underline{\mathbf{A}}_{j,:}^{(k)}(\mathbf{x}) + \underline{b}_j^{(k)}$ and $\bar{y}_j^{(k)}(\mathbf{x}) = \bar{\mathbf{A}}_{j,:}^{(k)}(\mathbf{x}) + \bar{b}_j^{(k)}$ have already been obtained when applying linear bound propagation for $B$. Computing $q(B; y_j^{(k)})$ is tractable, since it corresponds to the cumulative density function of a linear transformation of $\mathbf{X}$ at the origin. It should be noted that even if Equation (7) does not hold, one can still view the above defined $q(B; \cdot)$ as an approximation of $q(B_1)$ and $q(B_2)$ and use it as a guidance for selecting unstable preactivations to split.

Based on uncertainty level, we propose two strategies for selecting the preactivation to split for a given branch $B = \langle p_\ell, p_u, \mathcal{C} \rangle$.

**BaB-prob-ordered.** The first strategy is naïve, but turns out to be useful in many cases, especially in the verification of MLP models. We call BaB-prob with this naïve strategy BaB-prob-ordered. BaB-prob-ordered starts from the first ReLU layer and checks whether there is any unstable preactivation in this layer. If so, BaB-prob-ordered selects one such preactivation to split on; otherwise, BaB-prob-ordered proceeds to the next ReLU layer, until an unstable preactivation is found. $B$ must contain an unstable preactivation; otherwise, Proposition 4 implies $p_\ell = p_u$ and $B$ would not have been popped from $\mathcal{B}$. Since BaB-prob-ordered splits on an unstable preactivation $y_j^{(k)}$ only when there is no unstable preactivation in the first $k-1$ ReLU layers, it follows that $\underline{y}_j^{(k)}(\mathbf{x}) = \bar{y}_j^{(k)}(\mathbf{x})$, since no relaxation is required to bound $y_j^{(k)}$. Therefore, $q(B; y_j^{(k)}) = 0$. This explains why BaB-prob-ordered works well in many cases: BaB-prob-ordered always splits on the preactivation with uncertainty level of zero.

**BaB+BaBSR-prob.** Splitting on the unstable preactivation with uncertainty level of 0 does not necessarily minimize the gaps $p_{u,1} - p_{\ell,1}$ and $p_{u,2} - p_{\ell,2}$ since the uncertainty level is an upper bound on a lower bound on these gaps. Therefore, only considering uncertainty levels can sometimes be misguiding, making the algorithm inefficient, especially for CNN models. Actually, in image-classification scenarios, an MLP may contain hundreds or thousands of neurons in one activation layer, while a CNN can contain tens of thousands of neurons due to their channel and spatial dimensions. To resolve this issue, we propose to combine heuristics from deterministic verification with the uncertainty levels of preactivations to design splitting strategies. Specifically, we combine BaBSR (Bunel et al. (2020)), which estimates the improvement on the lower bound of $f(\mathbf{x})$ after splitting a preactivation, with the uncertainty levels of preactivations. We call this heuristic BaBSR-prob and call BaB-prob with BaBSR-prob heuristics BaB+BaBSR-prob.

BaB+BaBSR-prob first computes BaBSR scores for all the unstable preactivations (see Bunel et al. (2020) for details). It then enumerates the unstable preactivations in decreasing order of their BaBSR scores and evaluates the uncertainty level of each. If the uncertainty level of the current preactivation does not exceed a specified nonnegative threshold, BaB+BaBSR-prob splits on it; otherwise, it proceeds to the next preactivation with a lower BaBSR score. Note that there must exist at least one unstable preactivation in $B$ with uncertainty level below the threshold, since the one chosen by BaB-prob-ordered always has uncertainty level 0.

## 3.4 THEORETICAL RESULTS

In this section, we provide the theoretical properties of BaB-prob and defer their proofs to Appendix B.

**Proposition 1.** *The values of $p_\ell$ and $p_u$ computed in Equation (4) are lower and upper bounds on $\mathbb{P}(f(\mathbf{X}) > 0, \mathcal{C})$.*

**Proposition 2.** *For a given branch $B = \langle p_\ell, p_u, \mathcal{C} \rangle$, where $\mathcal{C} = \{y_{j_1}^{(k_1)} \geq 0, \ldots, y_{j_s}^{(k_s)} \geq 0, y_{j_{s+1}}^{(k_{s+1})} < 0, \ldots, y_{j_t}^{(k_t)} < 0, y_{j^\star}^{(k^\star)} \geq 0/y_{j^\star}^{(k^\star)} < 0\}$, Assume $\underline{f}(\mathbf{x})$, $\bar{f}(\mathbf{x})$ and $\underline{\mathbf{y}}^{[N]}(\mathbf{x})$, $\bar{\mathbf{y}}^{[N]}(\mathbf{x})$ are the linear lower and bounds obtained by linear bound propagation for $B$. Then,*

$$p_u - p_\ell \geq \mathbb{P}\begin{pmatrix} \bar{f}(\mathbf{X}) > 0, \ \bar{y}_{j^\star}^{(k^\star)}(\mathbf{X}) \geq 0, \ \underline{y}_{j^\star}^{(k^\star)}(\mathbf{X}) < 0, \\ \bar{y}_{j_1}^{(k_1)}(\mathbf{X}) \geq 0, \ldots, \bar{y}_{j_s}^{(k_s)}(\mathbf{X}) \geq 0, \ \underline{y}_{j_{s+1}}^{(k_{s+1})}(\mathbf{X}) < 0, \ldots, \underline{y}_{j_t}^{(k_{s+1})}(\mathbf{X}) < 0 \end{pmatrix}. \quad (9)$$

**Proposition 3.** *The global lower bound and upper bounds, denoted as $P_\ell$ and $P_u$, computed at the beginning of each iteration are indeed lower and upper bounds on $\mathbb{P}(f(\mathbf{X}) > 0)$.*

**Proposition 4.** *If $B = \langle p_\ell, p_u, \mathcal{C} \rangle$ has no unstable preactivation, then $\mathbb{P}(B) = p_\ell = p_u$.*

**Proposition 5.** *BaB-prob terminates in finite time.*

**Corollary 1.** *BaB-prob is both sound and complete.*

## 4 EXPERIMENTS

This section evaluates BaB-prob as well as other state-of-the-art probabilistic verifiers for *local probabilistic robustness*: for a correctly classified input $\mathbf{x}_0$, we add zero-mean Gaussian noise of small covariance and ask whether the network preserves the desired margin with probability at least $\eta$. We compare the two versions of BaB-prob—BaB-prob-ordered and BaB+BaBSR-prob—against PROVEN (Weng et al. (2019)), PV (Boetius et al.), and an SDP-based verifier (Fazlyab et al. (2019)), denoted as SDP. Because PROVEN's code is not public, we re-implemented it for consistency across baselines. We first check the soundness and completeness of the algorithms by conducting experiments on a toy MLP and a toy CNN. Results show that the verification results of both versions of BaB-prob and PV match the "ground truth," indicating that they are both sound and complete. PROVEN and SDP are shown incomplete, which is as expected. Details are provided in Appendix C.1. We then report results on (i) untrained MLP and CNN models across widths and depths, (ii) MNIST (LeCun (1998)) and CIFAR-10 (Krizhevsky et al. (2009)) models, respectively, and (iii) VNN-COMP 2025 benchmarks[1]. Unless stated otherwise, the desired probability threshold $\eta = 0.95$, the nonnegative threshold for BaBSR-prob is 0.01, and the per-instance time limit is 120 seconds. Computation is parallelized on GPU. Experiments were run on Ubuntu 22.04 with an Intel i9-14900K CPU, 64 GB RAM, and an RTX 4090 GPU, using a single CPU thread.

**Probability estimation and confidence.** In practive, analytically computing the probabilities in Equation (4) can be time-consuming when the underlying Gaussian distribution is high-dimensional. While dimensionality-reduction techniques could accelerate this computation, we approximate the probabilities using Monte Carlo sampling and certify the final decision with a statistical confidence guarantee. Across nearly all problems, our certificates achieve confidence exceeding $1 - 10^{-4}$. For completeness, Appendix C.4 details our confidence calculation and empirical confidence results for our approach.

**Success-rate definition.** For sound-and-complete methods, PV, BaB-prob-ordered and BaB+BaBSR-prob, "success rate" is the fraction of instances declared within the time limit. For PROVEN and SDP, we count a verification as successful only if it matches the declaration of any of PV or our two versions; in our runs there was no case where all three (PV and our two versions of BaB-prob) failed to declare while PROVEN or SDP succeeded.

In our experiments, SDP either exhausted memory or failed to solve the problems within the time limit. Therefore, we omit its results from the main text and report them in the appendix. Additional details about the experiments are provided in Appendix C. Our code is available at https://anonymous.4open.science/r/BaB-prob-59D3/.

### 4.1 UNTRAINED MLP AND CNN MODELS

We evaluate the scalability of different approaches with respect to network size on untrained MLPs and CNNs. For MLPs, we vary the input dimension $D_i$, hidden dimension $D_h$, and the number of hidden layers $N_h$. For CNNs, we vary the input shape $(1, W_i, H_i)$ with $W_i = H_i$, the number of hidden channels $C_h$, and the number of hidden layers $N_h$. Figure 2 shows the success rate results for the various approaches. The results show a clear advantage for both versions of BaB-prob, which consistently achieved higher success rates than PROVEN and PV. On MLP models, BaB-prob-ordered shows better scalability than BaB+BaBSR-prob, whereas on CNN models, BaB+BaBSR-prob scales better than BaB-prob-ordered.

### 4.2 MNIST AND CIFAR-10 MODELS

**Success rate and average time.** Table 1 shows the success rate and avergae time results for MNIST and CIFAR-10 models. On MLPs, both versions of BaB-prob strictly dominate PROVEN and PV. Besides, BaB-prob-ordered shows higher success rates than BaB+BaBSR-prob. On CNNs, both

---

[1]https://github.com/VNN-COMP/vnncomp2025_benchmarks.git

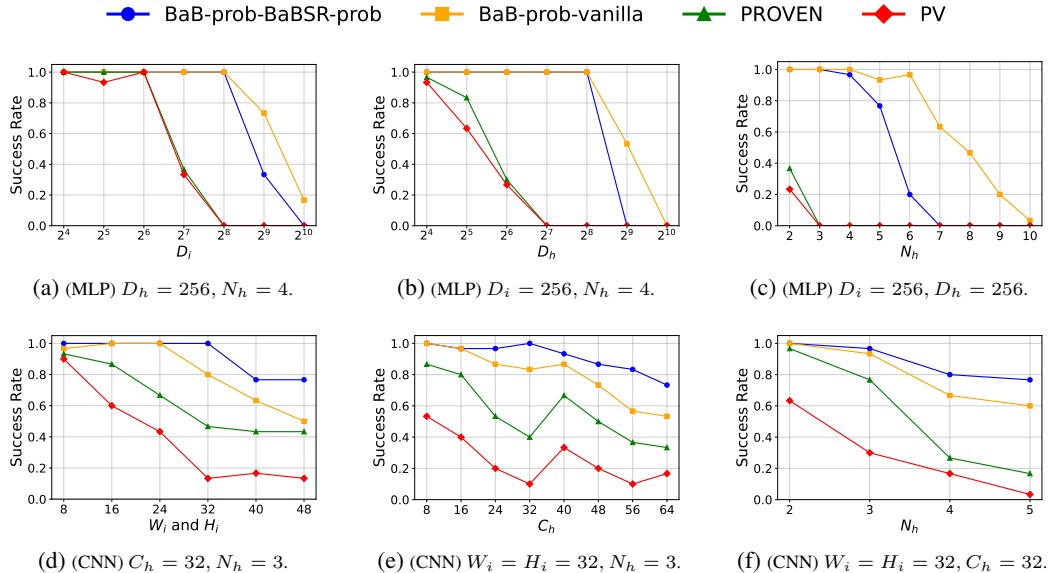

Figure 2: Success rate results for untrained models.

| MNIST models | | | | |
|---|---|---|---|---|
| | PROVEN | PV | BaB-prob-ordered (ours) | BaB+BaBSR-prob (ours) |
| MLP-256-2[1] | 70.00% (0.02s) | 33.33% (80.02s) | **100%** (0.25s) | **100%** (0.25s) |
| MLP-256-6 | 23.33% (0.02s) | 23.33% (92.01s) | **100%** (6.19s) | 86.67% (34.14s) |
| MLP-256-10 | 16.67% (0.04s) | 16.67% (100.01s) | 96.67% (11.61s) | 63.33% (51.80s) |
| MLP-1024-2 | 66.67% (0.01s) | 40.00% (72.02s) | **100%** (2.74s) | **100%** (1.21s) |
| MLP-1024-6 | 20.00% (0.03s) | 10.00% (108.01s) | **93.33%** (26.52s) | 56.67% (63.03s) |
| MLP-1024-10 | 6.67% (0.05s) | 6.67% (112.01s) | **56.67%** (65.01s) | 20.00% (102.73s) |
| Conv-8-2[2] | 96.67% (0.03s) | 20.00% (96.01s) | **100%** (0.03s) | **100%** (0.03s) |
| Conv-16-2 | 73.33% (0.02s) | 13.33% (104.01s) | 96.67% (13.33s) | **100%** (4.33s) |
| Conv-32-2 | 23.33% (0.02s) | 0% (120.03s) | 53.33% (65.61s) | **83.33%** (27.71s) |
| Conv-64-2 | 36.67% (0.02s) | 0% (120.04s) | 46.67% (69.87s) | **56.67%** (61.85s) |
| CIFAR-10 models | | | | |
| MLP-256-2 | 96.67% (0.03s) | 76.67% (28.02s) | **100%** (0.01s) | **100%** (0.02s) |
| MLP-256-6 | 60.00% (0.03s) | 53.33% (56.02s) | **100%** (0.32s) | **100%** (0.56s) |
| MLP-256-10 | 50.00% (0.04s) | 36.67% (76.02s) | **100%** (6.12s) | 86.67% (27.31s) |
| MLP-1024-2 | 93.33% (0.02s) | 90.00% (12.02s) | **100%** (0.02s) | **100%** (0.01s) |
| MLP-1024-6 | 66.67% (0.03s) | 60.00% (48.02s) | **100%** (3.02s) | 96.67% (8.77s) |
| MLP-1024-10 | 50.00% (0.05s) | 43.33% (68.02s) | **100%** (7.19s) | 83.33% (23.30s) |
| Conv-16-2 | 86.67% (0.03s) | 26.67% (88.02s) | 96.67% (4.50s) | **100%** (0.51s) |
| Conv-16-3 | 33.33% (0.03s) | 3.33% (116.02s) | 60.00% (56.02s) | **90.00%** (28.40s) |
| Conv-16-4 | 16.67% (0.04s) | 0% (120.02s) | 36.67% (81.57s) | **70.00%** (57.67s) |
| Conv-32-2 | 86.67% (0.03s) | 20.00% (96.03s) | 93.33% (8.95s) | **96.67%** (5.57s) |
| Conv-32-3 | 33.33% (0.04s) | 10.00% (108.03s) | 50.00% (69.85s) | **63.33%** (46.99s) |
| Conv-32-4 | 3.33% (0.05s) | 0.00% (120.04s) | 13.33% (107.64s) | **33.33%** (84.54s) |

[1] MLP-256-2 refers to the MLP model with $D_h = 256$ and $N_h = 2$.
[2] Conv-8-2 refers to the CNN model with $C_h = 8$ and $N_h = 2$.

Table 1: Success rate and average time results for MNIST and CIFAR-10 models.

versions of BaB-prob consistently outperform PROVEN and PV, and BaB+BaBSR-prob shows better performance than BaB-prob-ordered.

**Split efficiency.** Figure 3 compares the number of splits required by BaB-prob-ordered and BaB+BaBSR-prob. On MLPs, BaB-prob-ordered generally splits fewer preactivations than BaB+BaBSR-prob, particularly on the more challenging instances. For CNNs, BaB+BaBSR-

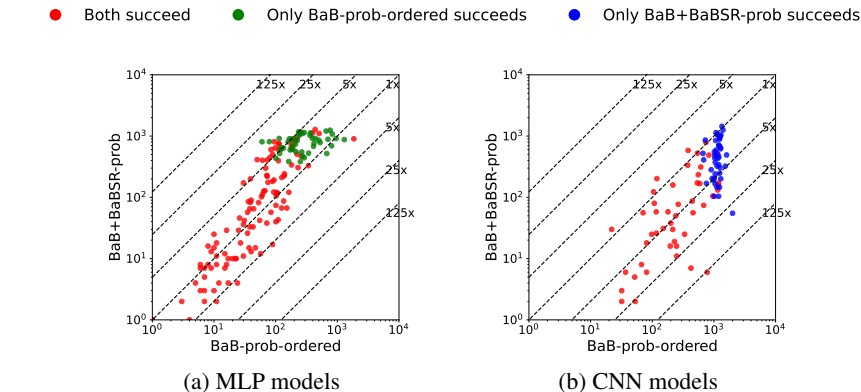

Figure 3: Comparison of number of splits between BaB-prob-ordered and BaB+BaBSR-prob.

| Dataset | Network Type | Input Dim | PROVEN | PV | BaB-prob-ordered (ours) | BaB+BaBSR-prob (ours) |
|---|---|---|---|---|---|---|
| acasxu_2023 | FC + ReLU | 5 | 10.22% (0.03s) | **92.47%** (15.09s) | 48.39% (63.83s) | 47.85% (65.59s) |
| cersyve | FC + ReLU (Control Tasks) | 2-5 | 0% (0.05s) | 91.67% (19.64s) | **100%** (8.88s) | **100%** (13.24s) |
| cifar100_2024 | FC, Conv, Residual + ReLU, BatchNorm | 3072 | 49.00% (0.17s) | 7.00% (111.69s) | 52.00% (59.78s) | **57.00%** (57.35%) |
| collins_rul_cnn_2022 | Conv + ReLU, Dropout | 400-800 | 91.94% (0.03s) | 93.55% (9.38s) | 95.16% (7.09s) | **96.77%** (5.07s) |
| cora_2024 | FC + ReLU | 784-3072 | 19.44% (0.04s) | 17.22% (99.34s) | **58.89%** (60.99s) | 34.44% (88.01s) |
| linearizenn_2024 | FC, Residual + ReLU | 4 | 20.00% (0.03s) | **100%** (0.08s) | 95.00% (21.46s) | 81.67% (42.37s) |
| relusplitter (MNIST) | FC+ReLU | 784 | 18.75% (0.02s) | 18.75% (97.51s) | **100%** (1.80s) | 96.25% (21.27s) |
| relusplitter (CIFAR-10) | Conv, Skip + ReLU, AvgPool | 3072 | 76.67% (0.04s) | 46.67% (64.02s) | 80.00% (24.71s) | **93.33%** (11.90s) |
| safenlp_2024 | FC + ReLU | 30 | 35.32% (0.01s) | 41.50% (73.82s) | 99.07% (3.18s) | **99.79%** (0.98s) |

Table 2: Results for VNN-COMP 2025 benchmarks.

prob consistently requires fewer preactivation splits than BaB-prob-ordered by up to two orders of magnitude.

### 4.3 RESULTS FOR VNN-COMP 2025 MODELS

Table 2 reports the results for the VNN-COMP 2025 benchmarks. Both versions of BaB-prob achieve higher success rates than PROVEN. Moreover, they consistently outperform PV, except on the low-dimensional datasets—acasxu_2023 (5 dimensions) and linearizenn_2024 (4 dimensions).

## 5 CONCLUSIONS

In this work, we have introduced BaB-prob, the first BaB framework with preactivation splitting for probabilistic verification of neural networks. BaB-prob iteratively divides the original problems into subproblems by splitting preactivations and leverages LiRPA to bound the probability for each subproblem. We prove soundness and completeness of our approach for ReLU networks, which can be extended to piecewise-linear activation functions. Furthermore, we introduce the notion of uncertainty level and propose two versions of BaB-prob with different strategies developed by uncertainty level. Extensive experiments show that our approach significantly outperforms state-of-the-art approaches in medium- to high-dimensional input problems.

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

---

**Algorithm 1** BaB-prob

---

**Input:** $f(\mathbf{x})$, $\mathbf{P}$, $\mathcal{D}$, $\eta$

1: $\mathcal{B} \leftarrow [\,]$

2: $\underline{f}(\mathbf{x}), \bar{f}(\mathbf{x}), \underline{\mathbf{y}}_o^{[N-1]}(\mathbf{x}), \bar{\mathbf{y}}_o^{[N-1]}(\mathbf{x}) \leftarrow ComputeLinearBounds(f, \mathcal{D}, \varnothing)$

3: $p_{\ell,o}, p_{u,o} \leftarrow BoundBranchProbability(\mathbf{P}, \underline{f}(\mathbf{x}), \bar{f}(\mathbf{x}), \underline{\mathbf{y}}_o^{[N-1]}(\mathbf{x}), \bar{\mathbf{y}}_o^{[N-1]}(\mathbf{x}), \varnothing)$

4: $B_o \leftarrow \langle p_{\ell,o}, p_{u,o}, \varnothing \rangle$

5: **if** $p_{\ell,o} < p_{u,o}$ **then**

6: $\quad MarkPreactivationToSplitOn(B_o)$

7: $\mathcal{B}.insert(B_o)$

8: **while True do**

9: $\quad P_\ell, P_u \leftarrow BoundGlobalProbability(\mathcal{B})$

10: $\quad$ **if** $P_\ell \geq \eta$ **then return** TRUE

11: $\quad$ **if** $P_u < \eta$ **then return** FALSE

12: $\quad B = \langle p_\ell, p_u, \mathcal{C} \rangle \leftarrow \mathcal{B}.pop()$

13: $\quad \{y_j^{(k)} \geq 0, y_j^{(k)} < 0\} \leftarrow GenerateNewConstraints(B)$

14: $\quad$ **for** $c_i \in \{y_j^{(k)} \geq 0, y_j^{(k)} < 0\}$ **do**

15: $\quad\quad \mathcal{C}_i \leftarrow \mathcal{C} \cup \{c_i\}$

16: $\quad\quad \underline{f}_i(\mathbf{x}), \bar{f}_i(\mathbf{x}), \underline{\mathbf{y}}_i^{[N-1]}(\mathbf{x}), \bar{\mathbf{y}}_i^{[N-1]}(\mathbf{x}) \leftarrow ComputeLinearBounds(f, \mathcal{D}, \mathcal{C}_i)$

17: $\quad\quad p_{\ell,i}, p_{u,i} \leftarrow BoundBranchProbability(\mathbf{P}, \underline{f}_i(\mathbf{x}), \bar{f}_i(\mathbf{x}), \underline{\mathbf{y}}_i^{[N-1]}(\mathbf{x}), \bar{\mathbf{y}}_i^{[N-1]}(\mathbf{x}), \mathcal{C}_i)$

18: $\quad\quad B_i \leftarrow \langle p_{l,i}, p_{u,i}, \mathcal{C}_i \rangle$

19: $\quad\quad$ **if** $p_{l,i} < p_{u,i}$ **then**

20: $\quad\quad\quad MarkPreactivationToSplitOn(B_i)$

21: $\quad\quad \mathcal{B}.insert(B_i)$

---

## A  DETAILED FRAMEWORK OF BaB-PROB

The pseudocode of BaB-prob is shown in Algorithm 1. During initialization, $\mathcal{B}$ is created to maintain all candidate branches (Line 1). BaB-prob applies linear bound propagation over $\mathcal{D}$ **under no constraint** to compute linear bounds for $f$ and $\mathbf{y}^{[N-1]}$, yielding $\underline{f}_o(\mathbf{x})$, $\bar{f}_o(\mathbf{x})$ and $\underline{\mathbf{y}}_o^{[N-1]}(\mathbf{x})$, $\bar{\mathbf{y}}_o^{[N-1]}(\mathbf{x})$, such that

$$\underline{f}_o(\mathbf{x}) \leq f(\mathbf{x}) \leq \bar{f}_o(\mathbf{x}), \qquad \forall \mathbf{x} \in \mathcal{D}, \tag{10a}$$

$$\underline{\mathbf{y}}_i^{(k)}(\mathbf{x}) \leq \mathbf{y}^{(k)}(\mathbf{x}) \leq \bar{\mathbf{y}}_o^{(k)}(\mathbf{x}), \qquad \forall \mathbf{x} \in \mathcal{D},\ k \in [N-1], \tag{10b}$$

(Line 2). BaB-prob then applies Equation (4) to compute the probability bounds for $\mathbb{P}\left(f(\mathbf{X}) > 0\right)$, obtaining the lower bound $p_{\ell,o}$ and the upper bound $p_{u,o}$, and creates the root branch $B_o = \langle p_{\ell,o}, p_{u,o}, \varnothing \rangle$ (Line 3-4). Before inserting $B_o$ into $\mathcal{B}$, the preactivation in $B_o$ to be split on is first identified (though not split immediately) (Line 5-6). This strategy reduces memory usage for BaB+BaBSR-prob, since the method relies on the linear bounds to choose the preactivation to split on. By marking the preactivation at this stage, we can discard the linear bound information before inserting $B_o$ into $\mathcal{B}$. Besides, BaB-prob only performs the identification if $(p_u - p_\ell)$ is positive. At the end of initialization, $B_o$ is inserted into $\mathcal{B}$ (Line 7).

At each iteration, BaB-prob first computes the global probability bounds for $\mathbb{P}\left(f(\mathbf{X}) > 0\right)$:

$$P_\ell = \sum_{\langle p_\ell, p_u, \mathcal{C} \rangle \in \mathcal{B}} p_\ell, \quad P_u = \sum_{\langle p_\ell, p_u, \mathcal{C} \rangle \in \mathcal{B}} p_u, \tag{11}$$

(Line 9). If the current global probability bounds are already enough to make a certification, BaB-prob terminates and make the corresponding certification (Line 10-11). Otherwise, BaB-prob pops the branch $B = \langle p_\ell, p_u, \mathcal{C} \rangle$ with the largest $(p_u - p_\ell)$ from $\mathcal{B}$ (Line 12) and generates the two new constraints on the identified preactivation (Line 13). For each new constraint $c_i$, BaB-prob generates a new set of constraints $\mathcal{C}_i = \mathcal{C} \cup \{c_i\}$ (Line 14-15). BaB-prob then applies linear bounds propagation over $\mathcal{D}$ **under the constraints in** $\mathcal{C}_i$ to compute linear bounds for $f$ and $\mathbf{y}^{[N-1]}$, yielding

$\underline{f}_i(\mathbf{x})$, $\bar{f}_i(\mathbf{x})$ and $\mathbf{y}_i^{[N-1]}(\mathbf{x})$, $\bar{\mathbf{y}}_i^{[N-1]}$, such that

$$\underline{f}_i(\mathbf{x}) \leq f(\mathbf{x}) \leq \bar{f}_i(\mathbf{x}), \qquad\qquad \forall \mathbf{x} \in \mathcal{D}, \ \mathcal{C}_i \text{ satisfied}, \tag{12a}$$

$$\mathbf{y}_i^{(k)}(\mathbf{x}) \leq \mathbf{y}^{(k)}(\mathbf{x}) \leq \bar{\mathbf{y}}_i^{(k)}(\mathbf{x}), \qquad \forall \mathbf{x} \in \mathcal{D}, \ \mathcal{C}_i^{[k-1]} \text{ satisfied}, \ k \in [N-1], \tag{12b}$$

(Line 16). Then, it uses Equation (4) to compute the probability bounds for $\mathbb{P}\left(f(\mathbf{X}) > 0, \mathcal{C}_i\right)$, obtaining $p_{\ell,i}$ and $p_{u,i}$. Same as the initialization, BaB-prob identifies the preactivation to split on for $B_i = \langle p_{\ell,i}, p_{u,i}, \mathcal{C}_i \rangle$ in advance (Line 19-20) and then inserts $B_i$ into $\mathcal{B}$.

# B  PROOF FOR THEORETICAL RESULTS

**Lemma 1.** *Let* $\mathcal{C} = \left\{ y_{j_\ell}^{k_\ell} \geq 0, \ell \in [s], y_{j_\ell}^{k_\ell}, \ell \in [s+1, t] \right\}$. *Assume* $\mathcal{C}$ *can be decomposed by*

$$\mathcal{C}^{(k)} = \left\{ y_{j_{k,\ell}}^{(k)} \geq 0, \ell \in [s_k], y_{j_{k,\ell}}^{(k)} < 0, \ell \in [s_k + 1, t_k] \right\}, \quad k \in [N-1]. \tag{13}$$

*Let and* $\underline{f}(\mathbf{x})$, $\bar{f}(\mathbf{x})$, $\mathbf{y}^{[N-1]}(\mathbf{x})$, $\bar{\mathbf{y}}^{[N-1]}(\mathbf{x})$ *be the linear bounds for* $f$ *and* $\mathbf{y}^{[N-1]}$ *obtained by linear bound propagation under the constraints of* $\mathcal{C}$. *For* $k \in [N-1]$, *let*

$$\mathcal{C}_x^{(k)} = \left\{ \mathbf{x} \in \mathcal{D} : y_{j_{k,\ell}}^{(k)}(\mathbf{x}) \geq 0, \ell \in [s_k], y_{j_{k,\ell}}^{(k)}(\mathbf{x}) < 0, \ell \in [s_k + 1, t_k] \right\},$$

$$\underline{\mathcal{C}}_x^{(k)} = \left\{ \mathbf{x} \in \mathcal{D} : \underline{y}_{j_{k,\ell}}^{(k)}(\mathbf{x}) \geq 0, \ell \in [s_k], \bar{y}_{j_{k,\ell}}^{(k)}(\mathbf{x}) < 0, \ell \in [s_k + 1, t_k] \right\}, \tag{14}$$

$$\bar{\mathcal{C}}_x^{(k)} = \left\{ \mathbf{x} \in \mathcal{D} : \bar{y}_{j_{k,\ell}}^{(k)}(\mathbf{x}) \geq 0, \ell \in [s_k], \underline{y}_{j_{k,\ell}}^{(k)}(\mathbf{x}) < 0, \ell \in [s_k + 1, t_k] \right\}.$$

*Then, for all* $k \in [N-1]$

$$\bigcap_{r=1}^{k} \underline{\mathcal{C}}_x^{(r)} \subseteq \bigcap_{r=1}^{k} \mathcal{C}_x^{(r)} \subseteq \bigcap_{r=1}^{k} \bar{\mathcal{C}}_x^{(r)}, \tag{15}$$

*and,*

$$\left( \bigcap_{r=1}^{N-1} \underline{\mathcal{C}}_x^{(r)} \right) \cap \{ \mathbf{x} \in \mathcal{D} : \underline{f}(\mathbf{x}) > 0 \}$$

$$\subseteq \left( \bigcap_{r=1}^{N-1} \mathcal{C}_x^{(r)} \right) \cap \{ \mathbf{x} \in \mathcal{D} : f(\mathbf{x}) > 0 \}$$

$$\subseteq \left( \bigcap_{r=1}^{N-1} \bar{\mathcal{C}}_x^{(r)} \right) \cap \{ \mathbf{x} \in \mathcal{D} : \bar{f}(\mathbf{x}) > 0 \}. \tag{16}$$

*Proof.* We first prove Equation (15) by induction.

By Equation (3b),

$$\underline{\mathbf{y}}^{(1)}(\mathbf{x}) \leq \mathbf{y}^{(1)}(\mathbf{x}) \leq \bar{\mathbf{y}}^{(1)}(\mathbf{x}), \quad \forall \mathbf{x} \in \mathcal{D}. \tag{17}$$

Therefore, $\underline{\mathcal{C}}_x^{(1)} \subseteq \mathcal{C}_x^{(1)} \subseteq \bar{\mathcal{C}}_x^{(1)}$, implying that Equation (15) holds for $k = 1$.

Assume Equation (15) holds for $k - 1$, we will prove Equation (15) for $k$.

$\forall \mathbf{x} \in \bigcap_{r=1}^{k} \underline{\mathcal{C}}_x^{(r)} \subseteq \bigcap_{r=1}^{k-1} \underline{\mathcal{C}}_x^{(r)}$, Equation (15) holding for $k - 1$ implies that $\mathbf{x} \in \bigcap_{r=1}^{k-1} \mathcal{C}_x^{(r)}$, i.e., $\mathcal{C}^{[k-1]}$ are satisfied. By Equation (3b),

$$\underline{\mathbf{y}}^{(k)}(\mathbf{x}) \leq \mathbf{y}^{(k)}(\mathbf{x}) \leq \bar{\mathbf{y}}^{(k)}(\mathbf{x}). \tag{18}$$

By Equation (18), and since $\mathbf{x} \in \underline{\mathcal{C}}_x^{(k)}$, it holds that $\mathbf{x} \in \mathcal{C}_x^{(k)}$. So we have $\mathbf{x} \in \bigcap_{r=1}^{k} \mathcal{C}_x^{(r)}$, implying that $\bigcap_{r=1}^{k} \underline{\mathcal{C}}_x^{(r)} \subseteq \bigcap_{r=1}^{k} \mathcal{C}_x^{(r)}$. For the second inequality, $\forall \mathbf{x} \in \bigcap_{r=1}^{k} \mathcal{C}_x^{(r)}$, Equation (18) also holds. Then, also, $\mathbf{x} \in \bar{\mathcal{C}}_x^{(k)}$. So $\mathbf{x} \in \bigcap_{r=1}^{k} \bar{\mathcal{C}}_x^{(r)}$, implying that $\bigcap_{r=1}^{k} \mathcal{C}_x^{(r)} \subseteq \bigcap_{r=1}^{k} \bar{\mathcal{C}}_x^{(r)}$. Thus, Equation (15) holds for $k$. Therefore, Equation (15) holds for all $k \in [N-1]$.

Using Equation (15) for $k = N - 1$, we can prove Equation (16) similar to how we prove Equation (15) from $k - 1$ to $k$. $\qquad\square$

*Proof of Proposition 1.* By Lemma 1 (Equation (16)),

$$\mathbb{P}\left(f(\mathbf{X}) > 0, \mathcal{C}\right) \geq \mathbb{P}\left(\underline{f}(\mathbf{X}) > 0, \underline{y}_{j_\ell}^{k_\ell}(\mathbf{X}) \geq 0, \ell \in [s], \bar{y}_{j_\ell}^{k_\ell}(\mathbf{X}) < 0, \ell \in [s+1, t]\right)$$
$$= \mathbb{P}\left(\underline{\mathbf{P}}\mathbf{X} + \underline{\mathbf{q}}\right). \tag{19}$$

Similarly, $\mathbb{P}\left(f(\mathbf{X}) > 0, \mathcal{C}\right) \leq \mathbb{P}\left(\bar{\mathbf{P}}\mathbf{X} + \bar{\mathbf{q}}\right).$ □

*Proof of Proposition 2.* We assume the constraint on $y_{j^\star}^{(k^\star)}$ in $\mathcal{C}$ is $y_{j^\star}^{(k^\star)} \geq 0$, the other case can be proved similarly. Assume that for $k \in [N-1]$, $k \neq k^\star$,

$$\mathcal{C}^{(k)} = \left\{ y_{j_{k,\ell}}^{(k)} \geq 0, \ell \in [s_k], \ y_{j_{k,\ell}}^{(k)} < 0, \ell \in [s_k+1, t_k] \right\}, \tag{20}$$

and

$$\mathcal{C}^{(k^\star)} = \left\{ y_{j_{k^\star,\ell}}^{(k^\star)} \geq 0, \ell \in [s_{k^\star}], \ y_{j_{k^\star,\ell}}^{(k^\star)} < 0, \ell \in [s_{k'}+1, t_{k^\star}], \ y_{j^\star}^{(k^\star)} \geq 0 \right\}. \tag{21}$$

By Proposition 1,

$$p_\ell = \mathbb{P}\left(\begin{matrix} \underline{f}(\mathbf{X}) > 0, \\ \underline{y}_{j_{k,\ell}}^{(k)}(\mathbf{X}) \geq 0, \ell \in [s_k], \qquad k \in [N-1], \\ \bar{y}_{j_{k,\ell}}^{(k)}(\mathbf{X}) < 0, \ell \in [s_k+1, t_k], \quad k \in [N-1] \end{matrix}\right). \tag{22}$$

Applying Lemma 1 (Equation (15)) with $k = N - 1$,

$$\bigcap_{r=1}^{N-1} \underline{\mathcal{C}}_x^{(r)} \subseteq \bigcap_{r=1}^{N-1} \mathcal{C}_x^{(r)}, \tag{23}$$

therefore,

$$\mathbb{P}\left(\begin{matrix} \underline{f}(\mathbf{X}) > 0, \\ \underline{y}_{j_{k,\ell}}^{(k)}(\mathbf{X}) \geq 0, \ell \in [s_k], \qquad k \in [N-1], \\ \bar{y}_{j_{k,\ell}}^{(k)}(\mathbf{X}) < 0, \ell \in [s_k+1, t_k], \quad k \in [N-1] \end{matrix}\right)$$

$$= \mathbb{P}\left(\begin{matrix} \underline{f}(\mathbf{X}) > 0, \\ \underline{y}_{j_{k,\ell}}^{(k)}(\mathbf{X}) \geq 0, \ell \in [s_k], \qquad k \in [N-1], \\ \bar{y}_{j_{k,\ell}}^{(k)}(\mathbf{X}) < 0, \ell \in [s_k+1, t_k], \quad k \in [N-1], \\ \mathcal{C}^{[N-1]} \end{matrix}\right)$$

$$= \mathbb{P}\left(\mathcal{C}^{[N-1]}\right) \mathbb{P}\left(\begin{matrix} \underline{f}(\mathbf{X}) > 0, \\ \underline{y}_{j_{k,\ell}}^{(k)}(\mathbf{X}) \geq 0, \ell \in [s_k], \qquad k \in [N-1], \\ \bar{y}_{j_{k,\ell}}^{(k)}(\mathbf{X}) < 0, \ell \in [s_k+1, t_k], \quad k \in [N-1] \end{matrix} \middle| \mathcal{C}^{[N-1]}\right). \tag{24}$$

By Equation (3a), we have

$$\mathbb{P}\left(\begin{matrix} \underline{f}(\mathbf{X}) > 0, \\ \underline{y}_{j_{k,\ell}}^{(k)}(\mathbf{X}) \geq 0, \ell \in [s_k], \qquad k \in [N-1], \\ \bar{y}_{j_{k,\ell}}^{(k)}(\mathbf{X}) < 0, \ell \in [s_k+1, t_k], \quad k \in [N-1] \end{matrix} \middle| \mathcal{C}^{[N-1]}\right)$$

$$\leq \mathbb{P}\left(\begin{matrix} \bar{f}(\mathbf{X}) > 0, \\ \underline{y}_{j_{k,\ell}}^{(k)}(\mathbf{X}) \geq 0, \ell \in [s_k], \qquad k \in [N-1], \\ \bar{y}_{j_{k,\ell}}^{(k)}(\mathbf{X}) < 0, \ell \in [s_k+1, t_k], \quad k \in [N-1] \end{matrix} \middle| \mathcal{C}^{[N-1]}\right). \tag{25}$$

From Equation (22) (24) (25),

$$
p_\ell \leq \mathbb{P}\left(\mathcal{C}^{[N-1]}\right) \mathbb{P}\left(\begin{array}{ll|} \bar{f}(\mathbf{X}) > 0, & \\ \underline{y}_{j_{k,\ell}}^{(k)}(\mathbf{X}) \geq 0,\, \ell \in [s_k], & k \in [N-1], \\ \bar{y}_{j_{k,\ell}}^{(k)}(\mathbf{X}) < 0,\, \ell \in [s_k+1, t_k], & k \in [N-1] \end{array}\,\middle|\, \mathcal{C}^{[N-1]}\right)
$$

$$
= \mathbb{P}\left(\begin{array}{ll} \bar{f}(\mathbf{X}) > 0, & \\ \underline{y}_{j_{k,\ell}}^{(k)}(\mathbf{X}) \geq 0,\, \ell \in [s_k], & k \in [N-1], \\ \bar{y}_{j_{k,\ell}}^{(k)}(\mathbf{X}) < 0,\, \ell \in [s_k+1, t_k], & k \in [N-1], \\ \mathcal{C}^{[N-1]} & \end{array}\right)
$$

$$
\leq \mathbb{P}\left(\begin{array}{ll} \bar{f}(\mathbf{X}) > 0, & \\ \underline{y}_{j_{k,\ell}}^{(k)}(\mathbf{X}) \geq 0,\, \ell \in [s_k], & k \in [N-1], \\ \bar{y}_{j_{k,\ell}}^{(k)}(\mathbf{X}) < 0,\, \ell \in [s_k+1, t_k], & k \in [N-1], \\ \mathcal{C}^{[N-2]} & \end{array}\right). \tag{26}
$$

Applying the argument of Equation (24)-(26), except that we apply Equation (3b) instead of Equation (3a) within Equation (25), iteratively to the constrained preactivations in layers $N-1, \ldots, k^*+1$, we obtain

$$
p_\ell \leq \mathbb{P}\left(\begin{array}{ll} \bar{f}(\mathbf{X}) > 0, & \\ \bar{y}_{j_{k,\ell}}^{(k)}(\mathbf{X}) \geq 0,\, \ell \in [s_k], & k \in [k^\star+1, N-1] \\ \underline{y}_{j_{k,\ell}}^{(k)}(\mathbf{X}) < 0,\, \ell \in [s_k+1, t_k], & k \in [k^\star+1, N-1] \\ \underline{y}_{j_{k,\ell}}^{(k)}(\mathbf{X}) \geq 0,\, \ell \in [s_k], & k \in [k^\star] \\ \bar{y}_{j_{k,\ell}}^{(k)}(\mathbf{X}) < 0,\, \ell \in [s_k+1, t_k], & k \in [k^\star] \\ \underline{y}_{j^\star}^{(k^\star)}(\mathbf{X}) \geq 0, & \\ \mathcal{C}^{[k^\star-1]} & \end{array}\right)
$$

$$
= \mathbb{P}\left(\mathcal{C}^{[k^\star-1]}\right) \mathbb{P}\left(\begin{array}{ll|} \bar{f}(\mathbf{X}) > 0, & \\ \bar{y}_{j_{k,\ell}}^{(k)}(\mathbf{X}) \geq 0,\, \ell \in [s_k], & k \in [k^\star+1, N-1] \\ \underline{y}_{j_{k,\ell}}^{(k)}(\mathbf{X}) < 0,\, \ell \in [s_k+1, t_k], & k \in [k^\star+1, N-1] \\ \underline{y}_{j_{k,\ell}}^{(k)}(\mathbf{X}) \geq 0,\, \ell \in [s_k], & k \in [k^\star] \\ \bar{y}_{j_{k,\ell}}^{(k)}(\mathbf{X}) < 0,\, \ell \in [s_k+1, t_k], & k \in [k^\star] \\ \underline{y}_{j^\star}^{(k^\star)}(\mathbf{X}) \geq 0 & \end{array}\,\middle|\, \mathcal{C}^{[k^\star-1]}\right). \tag{27}
$$

Applying Equation (3b) with $k = k^*$,

$$
\underline{y}_{j_{k^\star,\ell}}^{(k^\star)}(\mathbf{x}) \leq y_{j_{k^\star,\ell}}^{(k^\star)}(\mathbf{x}) \leq \bar{y}_{j_{k^\star,\ell}}^{(k^\star)}(\mathbf{x}), \qquad \forall \mathbf{x} \in \mathcal{D},\, \mathcal{C}^{[k^\star-1]} \text{ satisfied},\, \ell \in [t_{k^\star}], \tag{28a}
$$

$$
\underline{y}_{j^\star}^{(k^\star)}(\mathbf{x}) \leq y_{j^\star}^{(k^\star)}(\mathbf{x}) \leq \bar{y}_{j^\star}^{(k^\star)}(\mathbf{x}), \qquad \forall \mathbf{x} \in \mathcal{D},\, \mathcal{C}^{[k^\star-1]} \text{ satisfied}. \tag{28b}
$$

Thus,

$$
\begin{aligned}
&\mathbb{P}\left(\begin{array}{ll}
\bar{f}(\mathbf{X}) > 0, & \\
\bar{y}_{j_{k,\ell}}^{(k)}(\mathbf{X}) \geq 0, \ell \in [s_k], & k \in [k^\star + 1, N-1] \\
\underline{y}_{j_{k,\ell}}^{(k)}(\mathbf{X}) < 0, \ell \in [s_k + 1, t_k], & k \in [k^\star + 1, N-1] \\
\underline{y}_{j_{k,\ell}}^{(k)}(\mathbf{X}) \geq 0, \ell \in [s_k], & k \in [k^\star] \\
\bar{y}_{j_{k,\ell}}^{(k)}(\mathbf{X}) < 0, \ell \in [s_k + 1, t_k], & k \in [k^\star] \\
\underline{y}_{j^\star}^{(k^\star)}(\mathbf{X}) \geq 0 &
\end{array} \middle| \mathcal{C}^{[k^\star - 1]}\right) \\[2mm]
\leq\; &\mathbb{P}\left(\begin{array}{ll}
\bar{f}(\mathbf{X}) > 0, & \\
\bar{y}_{j_{k,\ell}}^{(k)}(\mathbf{X}) \geq 0, \ell \in [s_k], & k \in [k^\star, N-1] \\
\underline{y}_{j_{k,\ell}}^{(k)}(\mathbf{X}) < 0, \ell \in [s_k + 1, t_k], & k \in [k^\star, N-1] \\
\underline{y}_{j_{k,\ell}}^{(k)}(\mathbf{X}) \geq 0, \ell \in [s_k], & k \in [k^\star - 1] \\
\bar{y}_{j_{k,\ell}}^{(k)}(\mathbf{X}) < 0, \ell \in [s_k + 1, t_k], & k \in [k^\star - 1] \\
\underline{y}_{j^\star}^{(k^\star)}(\mathbf{X}) \geq 0 &
\end{array} \middle| \mathcal{C}^{[k^\star - 1]}\right) \\[2mm]
=\; &\mathbb{P}\left(\begin{array}{ll}
\bar{f}(\mathbf{X}) > 0, & \\
\bar{y}_{j_{k,\ell}}^{(k)}(\mathbf{X}) \geq 0, \ell \in [s_k], & k \in [k^\star, N-1] \\
\underline{y}_{j_{k,\ell}}^{(k)}(\mathbf{X}) < 0, \ell \in [s_k + 1, t_k], & k \in [k^\star, N-1] \\
\underline{y}_{j_{k,\ell}}^{(k)}(\mathbf{X}) \geq 0, \ell \in [s_k], & k \in [k^\star - 1] \\
\bar{y}_{j_{k,\ell}}^{(k)}(\mathbf{X}) < 0, \ell \in [s_k + 1, t_k], & k \in [k^\star - 1] \\
\underline{y}_{j^\star}^{(k^\star)}(\mathbf{X}) \geq 0, \bar{y}_{j^\star}^{(k^\star)}(\mathbf{X}) \geq 0 &
\end{array} \middle| \mathcal{C}^{[k^\star - 1]}\right).
\end{aligned}
\tag{29}
$$

From Equation (27) (29),

$$
\begin{aligned}
p_\ell \leq\; &\mathbb{P}\left(\mathcal{C}^{[k^\star - 1]}\right) \mathbb{P}\left(\begin{array}{ll}
\bar{f}(\mathbf{X}) > 0, & \\
\bar{y}_{j_{k,\ell}}^{(k)}(\mathbf{X}) \geq 0, \ell \in [s_k], & k \in [k^\star, N-1] \\
\underline{y}_{j_{k,\ell}}^{(k)}(\mathbf{X}) < 0, \ell \in [s_k + 1, t_k], & k \in [k^\star, N-1] \\
\underline{y}_{j_{k,\ell}}^{(k)}(\mathbf{X}) \geq 0, \ell \in [s_k], & k \in [k^\star - 1] \\
\bar{y}_{j_{k,\ell}}^{(k)}(\mathbf{X}) < 0, \ell \in [s_k + 1, t_k], & k \in [k^\star - 1] \\
\underline{y}_{j^\star}^{(k^\star)}(\mathbf{X}) \geq 0, \bar{y}_{j^\star}^{(k^\star)}(\mathbf{X}) \geq 0 &
\end{array} \middle| \mathcal{C}^{[k^\star - 1]}\right) \\[2mm]
=\; &\mathbb{P}\left(\begin{array}{ll}
\bar{f}(\mathbf{X}) > 0, & \\
\bar{y}_{j_{k,\ell}}^{(k)}(\mathbf{X}) \geq 0, \ell \in [s_k], & k \in [k^\star, N-1] \\
\underline{y}_{j_{k,\ell}}^{(k)}(\mathbf{X}) < 0, \ell \in [s_k + 1, t_k], & k \in [k^\star, N-1] \\
\underline{y}_{j_{k,\ell}}^{(k)}(\mathbf{X}) \geq 0, \ell \in [s_k], & k \in [k^\star - 1] \\
\bar{y}_{j_{k,\ell}}^{(k)}(\mathbf{X}) < 0, \ell \in [s_k + 1, t_k], & k \in [k^\star - 1] \\
\underline{y}_{j^\star}^{(k^\star)}(\mathbf{X}) \geq 0, \bar{y}_{j^\star}^{(k^\star)}(\mathbf{X}) \geq 0, & \\
\mathcal{C}^{[k^\star - 1]} &
\end{array}\right).
\end{aligned}
\tag{30}
$$

Then, again applying the argument of Equation (24)-(26), except that Equation (3b) is used instead of Equation (3a) within Equation (25), iteratively to the constrained preactivations in layers $k^\star -$

1, . . . , 1, we obtain

$$
p_\ell \leq \mathbb{P} \begin{pmatrix} \bar{f}(\mathbf{X}) > 0, & \\ \bar{y}_{j_{k,\ell}}^{(k)}(\mathbf{X}) \geq 0, \ell \in [s_k], & k \in [N-1] \\ \underline{y}_{j_{k,\ell}}^{(k)}(\mathbf{X}) < 0, \ell \in [s_k+1, t_k], & k \in [N-1] \\ \underline{y}_{j^\star}^{(k^\star)}(\mathbf{X}) \geq 0, \bar{y}_{j^\star}^{(k^\star)}(\mathbf{X}) \geq 0 & \end{pmatrix}
$$

$$
= \mathbb{P} \begin{pmatrix} \bar{f}(\mathbf{X}) > 0, \ \underline{y}_{j^\star}^{(k^\star)}(\mathbf{X}) \geq 0, \ \bar{y}_{j^\star}^{(k^\star)}(\mathbf{X}) \geq 0, \\ \bar{y}_{j_\ell}^{(k_\ell)}(\mathbf{X}) \geq 0, \ell \in [s], \ \underline{y}_{j_\ell}^{(k_\ell)}(\mathbf{X}) < 0, \ell \in [s+1, t] \end{pmatrix}. \tag{31}
$$

On the other hand, from Proposition 1,

$$
p_u = \mathbb{P} \begin{pmatrix} \bar{f}(\mathbf{X}) > 0, \ \bar{y}_{j^\star}^{(k^\star)}(\mathbf{X}) \geq 0, \\ \bar{y}_{j_\ell}^{(k_\ell)}(\mathbf{X}) \geq 0, \ell \in [s], \ \underline{y}_{j_\ell}^{(k_\ell)}(\mathbf{X}) < 0, \ell \in [s+1, t] \end{pmatrix}. \tag{32}
$$

Then, from Equation (31) (32),

$$
p_u - p_\ell \geq \mathbb{P} \begin{pmatrix} \bar{f}(\mathbf{X}) > 0, \ \bar{y}_{j^\star}^{(k^\star)}(\mathbf{X}) \geq 0, \ \underline{y}_{j^\star}^{(k^\star)}(\mathbf{X}) < 0, \\ \bar{y}_{j_\ell}^{(k_\ell)}(\mathbf{X}) \geq 0, \ell \in [s], \ \underline{y}_{j_\ell}^{(k_\ell)}(\mathbf{X}) < 0, \ell \in [s+1, t] \end{pmatrix} \tag{33}
$$

$\square$

*Proof of Proposition 3.* Notice that after BaB-prob splits on a preactivation in $B$ and generates $B_1$ and $B_2$, it holds that

$$
\mathbb{P}(B) = \mathbb{P}(B_1) + \mathbb{P}(B_2). \tag{34}
$$

Thus, at the beginning each iteration,

$$
\mathbb{P}(f(\mathbf{X}) > 0) = \sum_{B \in \mathcal{B}} \mathbb{P}(B). \tag{35}
$$

By Proposition 1, for all $B = \langle p_\ell, p_u, \mathcal{C} \rangle \in \mathcal{B}$, $p_\ell$ and $p_u$ are lower bounds on $\mathbb{P}(B)$. Therefore, from Equation (35), $P_\ell$ and $P_u$ are lower and upper bounds on $\mathbb{P}(f(\mathbf{X}) > 0)$. $\square$

*Proof of Proposition 4.* Let $\underline{f}(\mathbf{x})$, $\bar{f}(\mathbf{x})$ and $\underline{\mathbf{y}}^{[N-1]}(\mathbf{x})$, $\bar{\mathbf{y}}^{[N-1]}(\mathbf{x})$ be the linear lower and upper bounds for $f$ and $\mathbf{y}^{[N-1]}$ computed by linear bound propagation under the constraints in $\mathcal{C}$. Since there is no unstable preactivation in $B$, ino relaxation is performed during the linear bound propagation, the inequalities in Equation (3) become equalities. Thus, all the inclusions '$\subseteq$' in the proof of Lemma 1 become equalities '$=$', and subsequently, the inequality in the proof of Proposition 1 becomes equality. Therefore, $\mathbb{P}(B) = p_\ell = p_u$. $\square$

*Proof of Proposition 5.* We prove Proposition 5 by contradiction. Suppose that BaB-prob does not terminate in finite time. Since there are only finitely many preactivations, there must exist a point at which every branch in $\mathcal{B}$ contains no unstable preactivation. At that point, by Proposition 4, all branches have exactly tight probability bounds. Consequently, $P_\ell = P_u$, which implies that BaB-prob has already terminated — a contradiction. $\square$

*Proof of Corollary 1.* It follows directly from Proposition 3 and Proposition 5. $\square$

## C  EXPERIMENTS DETAILS

### C.1  TOY MODELS (SOUNDNESS AND COMPLETENESS CHECK)

**Setup.** The MLP model has input dimension 5, two hidden FC layers (10 units each), ReLU after every hidden layer, and a scalar output. The CNN model takes a $1 \times 4 \times 4$ input, has two

| | MLP model | | | | | | CNN model | | | | | |
|---|---|---|---|---|---|---|---|---|---|---|---|---|
| | T/T | F/T | N/F | T/F | F/F | N/F | T/T | F/T | N/F | T/F | F/F | N/F |
| PROVEN | 4/24 | 0/24 | 20/24 | 0/6 | 0/6 | 6/6 | 5/25 | 0/25 | 20/25 | 0/5 | 0/5 | 5/5 |
| PV | 24/24 | 0/24 | 0/24 | 0/6 | 6/6 | 0/6 | 25/25 | 0/25 | 0/25 | 0/5 | 5/5 | 0/5 |
| SDP | 0/24 | 24/24 | 0/24 | 0/6 | 6/6 | 0/6 | 0/25 | 25/25 | 0/25 | 0/5 | 5/5 | 0/5 |
| BaB-prob-ordered | 24/24 | 0/24 | 0/24 | 0/6 | 6/6 | 0/6 | 25/25 | 0/25 | 0/25 | 0/5 | 5/5 | 0/5 |
| BaB+BaBSR-prob | 24/24 | 0/24 | 0/24 | 0/6 | 6/6 | 0/6 | 25/25 | 0/25 | 0/25 | 0/5 | 5/5 | 0/5 |

Table 3: Results for toy models. Each cell indicates the number of instances where the algorithm's declaration ("T" for True, "F" for False, "N" for no declaration) aligns with the ground truth ("T" or "F"). For example, "T/T" means the ground truth is True and the algorithm declares it as True; "F/T" means the ground truth is True but the algorithm declares it as False; "N/T" means the ground truth is True but the algorithm fails to provide a declaration. Similar interpretations apply to the other entries.

Conv2d layers (3 channels, kernel 3, stride 2), ReLU after convolutions, then flatten + scalar FC. All weights/biases are i.i.d. $\mathcal{N}(0, 0.25)$. For each architecture we generate 30 instances, i.e., 30 different networks, such that $\mathbf{x}_0 = 0$ and $f(\mathbf{x}_0) > 0$. Noise is $\mathcal{N}(0, 0.1\mathbf{I})$. Because PROVEN, PV and BaB-prob assume bounded inputs, we truncate the Gaussian to its 99.7 %-confidence ball. We use batch size of 16384 for both versions of BaB-prob and PV, and split depth (maybe splitting multiple preactivations at one time) of 1 for this experiment. To verify the soundness and completeness, we do not set a time limit in this experiment. We use a toy MLP model and a toy CNN model to test the soundness and completeness of different solvers. We use SaVer-Toolbox[2] (Sivaramakrishnan et al. (2024)) to obtain an empirical reference $\hat{P}$ for $\mathbb{P}(f(\mathbf{X}) > 0)$ such that the deviation is $< 0.1\%$ with confidence $> 1 - 10^{-4}$. We treat the declaration of SaVer-Toolbox as the ground truth.

**Results.** Table 3 reports the counts of (declared TRUE/FALSE/No-declaration) vs. ground truth. Both BaB-prob versions and PV match ground truth on all toy problems; PROVEN returns only a subset (sound but incomplete), and SDP is mixed. It should be noted that this does not indicate that SDP is unsound, because it does not declare a TRUE problem as FALSE.

## C.2 OTHER EXPERIMENT SETUPS

**Untrained models.** The architecture for the untrained MLP models are same as that for the toy MLP model with the difference that the number of input features $D_i$, the number of hidden features $D_h$, and the number of hidden layers $N_h$ are not fixed. The architecture for the untrained CNN models are same as that for the toy CNN model with the difference that the input shape $(1, W_i, H_i)$ with $W_i = H_i$, the number of hidden channels $C_h$, and the number of hidden layers $N_h$ are not fixed. The weights and biases of all layers in the MLP and CNN models are also randomized with Gaussian distribution $\mathcal{N}(0, 0.25)$. We test on different combination of $(D_i, D_h, N_h)$ for MLP models and $(W_i/H_i, C_h, N_h)$ for CNN models. For each combination, we generated 30 different problems, each consists of a sample $\mathbf{x}_0 = 0$ and a randomly generated network $f$ such that $f(\mathbf{x}_0) > 0$. The noise is zero-mean with diagonal covariance, chosen so that its 99.7%-confidence ball has radius 0.002 for MLP models and 0.01 for CNN models. We use batch size of 4 for BaB-prob and PV, and split depth of 1 for this experiment.

**MNIST and CIFAR-10 models.** The MNIST and CIFAR-10 models have similar architecture as untrained models with the difference that the number of input features (or input shape) is fixed, the number of output features is 10, and the trained CNN models have a BatchNorm2d layer before each ReLU layer. We trained MLP models with different combination of $(D_h, N_h)$ and CNN models with different combination of $(C_h, N_h)$. The models are trained with cross-entropy loss and Adam optimizer. The batch size for training is 64. The learning rate is 0.001. We train each model for 20 iterations and use the checkpoint with the lowest loss on validation dataset for verification. For each model, we randomly selected 30 correctly classified samples from the training set. The noise is zero-mean with diagonal covariance, chosen so that its 99.7%-confidence ball has radius 0.02 for MNIST models and 0.01 for CIFAR-10 models. The output specification for each sample is $\mathcal{Y} = \{\mathbf{y} \in \mathbb{R}^{10} : (\mathbf{e}_t - \mathbf{e}_a)^{\mathsf{T}}\mathbf{y} > 0\}$, where $\mathbf{e}_t$ and $\mathbf{e}_a$ are the standard basis vectors, $t$ is the index

[2]https://github.com/vigsiv/SaVer-Toolbox

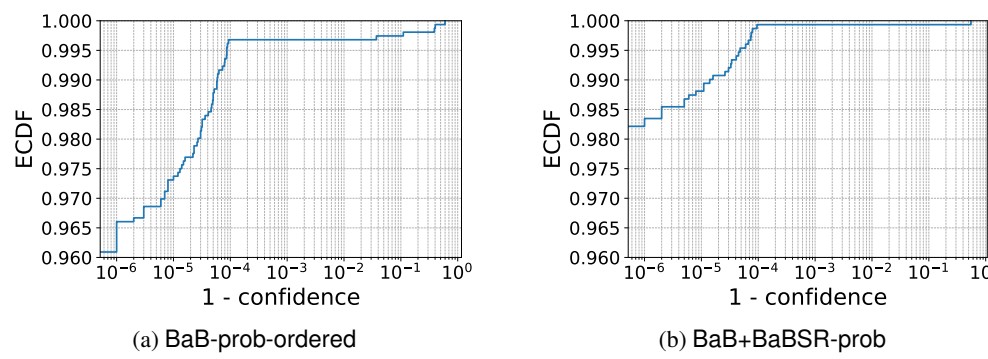

Figure 4: ECDF of confidence for BaB-prob-ordered and BaB+BaBSR-prob.

of the ground-truth label and $a \neq t$ is a randomly selected attacking label. $f(\mathbf{x}) \in \mathcal{Y}$ indicates that $\mathbf{x}$ is not misclassified by index $a$. We use batch size of 4 for BaB-prob and PV, and split depth of 1 for this experiment.

**VNN-COMP 2025 benchmarks.** We conducted evaluations on the following benchmark suites: `acasxu_2023`, `cersyve`, `cifar100_2024`, `collins_rul_cnn_2022`, `cora_2024`, `linearizenn_2024`, `relusplitter`, and `safenlp_2024`. Due to GPU memory limitations, for `cifar100_2024` we evaluated only the medium models. For `relusplitter`, we tested all MNIST models but only the oval21 models among the CIFAR-10 models. Models from other benchmark datasets were excluded either because they contained layer types not supported by our method or were too large to fit within GPU memory. For simplicity, we randomly selected one input region and one output specification from each original problem. The noise distribution is zero-mean with diagonal covariance, scaled such that its 99.7%-confidence ellipsoid matches the axis-aligned radii given in the original problem. In terms of solver configuration, we used a batch size of 8 for both BaB-prob and PV, with a split depth of 2. For the `cifar100_2024` benchmark, the batch size and split depth were set to 1 for BaB-prob, and the batch size was set to 4 for PV. Furthermore, since the original radii in `cifar100_2024` were too small to present a meaningful verification challenge, we doubled their values.

## C.3 SDP EXPERIMENTAL RESULTS

**Untrained models.** SDP failed to solve any MLP problem within the time limit and ran out of RAM on the CNN problems.

**MNIST and CIFAR-10 models.** SDP failed to solve any MLP problem within the time limit and ran out of RAM on the CNN problems.

**VNN-COMP 2025 benchmarks.** The SDP solver does not directly support `cersyve`, `cifar100_2024`, `linearizenn_2024`, `relusplitter` (CIFAR-10). Besides, it ran out of RAM on `cora_2024`, `collins_rul_cnn_2022` and `relusplitter` (MNIST). On `acasxu_2023` and `safenlp_2024`, it hit the time limit on all the problems.

## C.4 CONFIDENCE OF BAB-PROB

### Derivation of confidence

In our experiments, BaB-prob evaluates the per-branch probability in Equation 4 by Monte Carlo sampling, using $N = 10^5$ i.i.d. samples for each probability it needs to compute. Let $P_\ell$ and $P_u$ be the true global lower and upper probability bounds, and let $\hat{P}_\ell$ and $\hat{P}_u$ be their empirical values. When BaB-prob terminates, either $\hat{P}_\ell \geq \eta$ or $\hat{P}_u < \eta$. Then, the following proposition gives the confidence for the certification.

**Proposition 6.**

$$\mathbb{P}\left(P_\ell \geq \eta\right) \geq 1 - \exp\left(-\frac{N(\hat{P}_\ell - \eta)^2}{2V_1 + \frac{2}{3}(\hat{P}_\ell - \eta)}\right), \quad \text{if } \hat{P}_\ell \geq \eta; \tag{36a}$$

$$\mathbb{P}\left(P_u < \eta\right) \geq 1 - \exp\left(-\frac{N(\eta - \hat{P}_u)^2}{2V_2 + \frac{2}{3}(\eta - \hat{P}_u)}\right), \quad \text{if } \hat{P}_u < \eta. \tag{36b}$$

*where*

$$V_1 = \sum_{\langle p_\ell, p_u, \mathcal{C}\rangle \in \mathcal{B}} p_\ell(1 - p_\ell), \tag{37a}$$

$$V_2 = \sum_{\langle p_\ell, p_u, \mathcal{C}\rangle \in \mathcal{B}} p_u(1 - p_u). \tag{37b}$$

*Proof.* We prove the case of $\hat{P}_\ell \geq \eta$, and $\hat{P}_u < \eta$ can be proved similarly.

Denote $p_B := p_\ell$ for $B = \langle p_\ell, p_u, \mathcal{C}\rangle \in \mathcal{B}$. Let $\hat{p}_B$ be the empirical estimation for $p_B$ by Monte Carlo Sampling. Then,

$$\hat{p}_B = \frac{1}{N}\sum_{i=1}^{N} Z_{B,i}, \quad Z_{B,i} \sim \text{Bernoulli}(p_B), \tag{38}$$

and

$$P_\ell = \sum_{B \in \mathcal{B}} p_B, \quad \hat{P}_\ell = \sum_{B \in \mathcal{B}} \hat{p}_B. \tag{39}$$

Consider the estimation error

$$\hat{P}_\ell - P_\ell = \sum_{B \in \mathcal{B}}\sum_{i=1}^{N} \frac{1}{N}(Z_{B,i} - p_B). \tag{40}$$

The summands are independent, mean-zero, and bounded in $[-\frac{1}{N}, \frac{1}{N}]$; their total variance is

$$\text{Var}\left(\sum_{B \in \mathcal{B}}\sum_{i=1}^{N} \frac{1}{N}(Z_{B,i} - p_B)\right) = \frac{1}{N^2}\sum_{B \in \mathcal{B}}\sum_{i=1}^{N} p_B(1 - p_B) = \frac{1}{N}V_1. \tag{41}$$

Applying Bernstein's inequality with $\varepsilon = \hat{P}_\ell - \eta$ (Boucheron et al. (2013)),

$$\mathbb{P}\left(\hat{P}_\ell - P_\ell \geq \varepsilon\right) \leq \exp\left(-\frac{\frac{1}{2}\varepsilon^2}{\frac{1}{N}V_1 + \frac{1}{3N}\varepsilon}\right) = \exp\left(-\frac{N\varepsilon^2}{2V_1 + \frac{2}{3}\varepsilon}\right). \tag{42}$$

Therefore,

$$\mathbb{P}\left(P_\ell \geq \eta\right) \geq 1 - \exp\left(-\frac{N\varepsilon^2}{2V_1 + \frac{2}{3}\varepsilon}\right) \tag{43}$$

$\square$

The true values of $p_\ell$ and $p_u$ in Equation 37 are not directly accessible, so we use their empirical results as replacement. In our experiments, if BaB-prob-ordered or BaB+BaBSR-prob produces a declaration, that is, $\hat{P}_l \geq \eta$ or $\hat{P}_u < \eta$, but with confidence below $1 - 10^{-4}$, the algorithm continues running until the confidence reaches $1 - 10^{-4}$ or the time limit is hit. If when the algorithm terminates with a declaration but with confidence remaining below $1 - 10^{-4}$, we still count it a successful verification. The following results provide a statistical characterization of the achieved confidence levels.

**Confidence results**

Figure 4 presents the Empirical Cumulative Distribution Function (ECDF) of confidence values for BaB-prob-ordered and BaB+BaBSR-prob across all successfully verified problems. Both methods achieve confidence greater than $1 - 10^{-4}$ in over 99.5% of cases. This demonstrates that, in practice, the vast majority of problems are certified with very high confidence by both BaB-prob-ordered and BaB+BaBSR-prob.

## D   LLM USAGE

The authors acknowledge the use of GPT-4 and GPT-5 for polishing the main text.

