# OpenReview forum: "BaB-prob: Branch and Bound with Preactivation Splitting for Probabilistic Verification of Neural Networks"
_ICLR.cc/2026/Conference — ICLR 2026 Conference Withdrawn Submission_

### Official Review · Reviewer_J25x · 2025-10-23

**Soundness:** 1
**Presentation:** 4
**Contribution:** 2
**Rating:** 2
**Confidence:** 4

**Summary:**

This paper presents BaB-prob, an algorithm for verifying probabilistic specifications of neural networks. It combines ReLU splitting with Monte Carlo sampling to estimate event probabilities. The method includes heuristics for selecting ReLUs to split and is evaluated against existing probabilistic verification approaches.

**Strengths:**

- The paper explores the promising concept of applying neuron splitting to the probabilistic verification of neural networks.
- BaB-prob shows improved scalability over existing methods, though it may produce unsound results due to Monte Carlo sampling.
- The experimental evaluation includes a diverse set of challenging benchmarks.
- Theoretical analysis of BaB-prob yields several notable findings.
- The confidence interval analysis is valuable.
- The writing is concise and clear.

**Weaknesses:**

1. Zhang et al. already covered related ground in their TACAS 2024 paper, "Provable Preimage Under-Approximation for Neural Networks." Their work already introduces neuron splitting in the context of probabilistic verification, and they leverage both linear bound propagation and Monte Carlo sampling for probability estimation.

2. BaB-prob doesn't account for infeasible branches that arise during neuron splitting, which means the algorithm lacks completeness. The issue traces back to Proposition 4—the proof there has a flaw that undermines the completeness claim. Specifically, the assertion that "since there is no unstable preactivation in B, no relaxation is performed during the linear bound propagation, the inequalities in Equation (3) become equalities" breaks down when you're dealing with infeasible branches.

3. The theoretical results do not account for the Monte Carlo approximations that are required for practically applying this algorithm.

4. The description of how confidence intervals are handled in Appendix C.4 explains that when BaB-prob-ordered or BaB+BaBSR-prob produces a declaration with confidence below $1 - 10^{−4}$, the algorithm keeps running until either hitting that confidence threshold or reaching the time limit. The issue is that this approach actually invalidates the confidence levels—the problem is that the number of iterations becomes dependent on the confidence level itself, which breaks the statistical guarantees. Budde et al. have a nice paper in TACAS 2025 called "Sound Statistical Model Checking for Probabilities and Expected Rewards" that describes this issue in detail.

5. There's a fundamental issue with how the comparison is set up. The paper is essentially comparing Monte Carlo approximations—which give you potentially unsound results with confidence intervals—against algorithms that compute exact probabilities. That's not really an apples-to-apples comparison. The core problem is that once we allow potentially unsound results with confidence intervals, we're actually reducing the complexity of probabilistic verification.

**Questions:**

**Q1.** Please comment on the theoretical issues described above.
**Q2.** Can you compare your approach to Zhang et al. (2024) both conceptually and experimentally?
**Q3.** Can you conduct experiments where you compute exact probabilities for an apples-to-apples comparison?

---

### Official Review · Reviewer_FdAL · 2025-10-25

**Soundness:** 3
**Presentation:** 3
**Contribution:** 3
**Rating:** 6
**Confidence:** 4

**Summary:**

This paper presents BaB-prob, a branch-and-bound (BaB) framework for probabilistic verification of neural networks. It adapts preactivation-splitting—common in deterministic verification—to probabilistic settings by integrating linear bound propagation to estimate probability bounds for each branch. Two splitting heuristics are proposed: BaB-prob-ordered (simple layer-first rule) and BaB+BaBSR-prob (combining BaBSR with an “uncertainty level” metric). The method is proven sound and complete for ReLU networks and evaluated on untrained models, MNIST, CIFAR-10, and VNN-COMP 2025 benchmarks, showing improved scalability over PROVEN, PV, and SDP.

**Strengths:**

1. Extends a well-known deterministic BaB technique into the probabilistic domain with clear theoretical guarantees (soundness, completeness, termination).

2. Demonstrates consistent empirical gains over PROVEN and PV across several datasets and network types.

3. Sound implementation and evaluation; code is promised for release.

4. Solid integration of linear bound propagation for probability bounds.

**Weaknesses:**

1. Incremental novelty: The extension from deterministic to probabilistic BaB is conceptually straightforward; most components (bound propagation, splitting logic) are inherited from prior work.

2. Weak heuristic motivation: The “uncertainty level” is introduced heuristically with limited theoretical grounding. It is unclear how it generalizes or why it outperforms simple ordering beyond empirical evidence.

3. Scalability trade-offs: Monte Carlo probability estimation can become the bottleneck for high-dimensional Gaussian inputs, limiting real-world applicability.

**Questions:**

A recent work, “Towards Reliable Neural Specifications” (Geng et al., ICML 2023), proposed using neural activation patterns as verification specifications. Could BaB-prob be extended or adapted to handle such neural-specification–based properties, rather than standard input–output constraints?

---

### Official Review · Reviewer_2gZ9 · 2025-10-28

**Soundness:** 4
**Presentation:** 3
**Contribution:** 2
**Rating:** 4
**Confidence:** 3

**Summary:**

This paper presents a new sound and complete approach for proving probabilistic properties of neural networks. The work builds on previous work for proving deterministic properties that combines branch-and-bound techniques with linear bound propagation. The authors present a similar branch-and-bound approach for verifying probabilistic properties. The main adaptation required to prove probabilistic properties is the introduction of new splitting heuristics. The authors introduce two splitting heuristics based on an idea of uncertainty levels. The authors compare their method to multiple baselines on a variety of benchmarks and show improved performance.

**Strengths:**

- The paper gives adequate attention to all parts of the research including previous work, methods, and experiments.
- The algorithm provided is both sound and complete, so it can be used for exact probabilistic verification.
- The authors perform extensive experiments and compare their results to relevant baselines. These experiments show how the method scales to different network sizes and architectures and highlight the different behavior of their proposed splitting heuristics.
- The results significantly outperform the baseline methods.

**Weaknesses:**

- The method requires the user to assume that the input is bounded, so it must truncate input distributions that are defined for all real numbers.
- As the authors mention, equation (4) might be difficult to evaluate analytically, especially if the input distribution is not Gaussian. Propagating non-Gaussian distributions is a nontrivial task.
- The main contribution of the paper appears to be the splitting heuristics since the core of the method relies on standard bound propagation techniques.
- The splitting heuristics perform quite differently on MLPs vs CNNs, but no explanation or intuition is provided.

**Questions:**

- What is the intuition behind the different performance of the splitting heuristics on the CNN and MLP models?

Suggestions
- Line 28 typo: “asks whether a given satisfies”
- Line 77: function -> functions
- Font in figure 1 is a bit small (especially in the green boxes)
- Line 137: upper functions -> upper bound functions
- Line 138: add some description explaining why it is not required (I assume it is because ReLU is linear in this case)
- Line 345: practive -> practice
- Line 374: avergae -> average
- Line 470: the first time LiRPA is mentioned is in the conclusion, it would be good to mention earlier in the methods section

---

### Official Review · Reviewer_z8V7 · 2025-11-01

**Soundness:** 3
**Presentation:** 3
**Contribution:** 2
**Rating:** 4
**Confidence:** 4

**Summary:**

The authors extend branch-and-bound with preactivaction splitting to the setting of probabilistic verification of neural networks and demonstrate the soundness and completeness of their method for Linear/ReLU networks. The authors introduce two variants for selecting the preactivation to split based on their notion of uncertainty level, and benchmark them against competing methods.

**Strengths:**

* The main strength is extending the branch-and-bound framework with preactivation splitting to a new domain, that of probabilistic verification.
* The authors prove some useful theoretical properties of their approach
* The usefulness of the method is supported by strong empirical evidence
* Clear presentation, well-written paper

**Weaknesses:**

* The verifier is presented as sound and complete; however, in practice, results rely on Monte-Carlo sampling and certifying the results with a high confidence level.
* The novelty is limited. The paper is mainly a combination of widely-used techniques, like linear bound propagation, branch-and-bound, and probability estimation over linear regions
* Experiments focus on one kind of distribution: perturbations of input points with Gaussian noise. How would the method perform on other input distributions?
* No limitations/discussions paragraph/section included

**Questions:**

In addition to the above question, can you explain why the method seems to underperform on lower-dimensional datasets?

---

### Note · Authors · 2025-12-03

I have read and agree with the venue's withdrawal policy on behalf of myself and my co-authors.